



# Estimating daily full-coverage and high-accuracy 5-km ambient particulate matters across China: considering their precursors and chemical compositions

Yuan Wang[1], Qiangqiang Yuan[1,4,5], Tongwen Li[2], Siyu Tan[1], Liangpei Zhang[3,5]

[1]School of Geodesy and Geomatics, Wuhan University, Wuhan, Hubei, 430079, China.

[2]School of Geospatial Engineering and Science, Sun Yat-sen University, Zhuhai, Guangdong, 519082, China.

[3]The State Key Laboratory of Information Engineering in Surveying, Mapping and Remote Sensing, Wuhan University, Wuhan, Hubei, 430079, China.

[4]The Key Laboratory of Geospace Environment and Geodesy, Ministry of Education, Wuhan University, Wuhan, Hubei, 430079, China.

[5]The Collaborative Innovation Center for Geospatial Technology, Wuhan, Hubei, 430079, China.

*Correspondence to*: Qiangqiang Yuan (yqiang86@gmail.com)

**Abstract.** The ambient concentrations of particulate matters ($PM_{2.5}$ and $PM_{10}$) are significant indicators for monitoring the air quality relevant to living conditions. Most of the existing approaches for the estimation of $PM_{2.5}$ and $PM_{10}$ employed the remote sensing Aerosol Optical Depth (AOD) products as the main variate. Nevertheless, the coverage of missing data is generally large in AOD products, which can cause inconvenience to the researchers. To efficiently address this issue, our study explores a novel approach using the datasets of the precursors & chemical compositions for $PM_{2.5}$ and $PM_{10}$ instead of AOD products. Specifically, the daily full-coverage ambient concentrations of $PM_{2.5}$ and $PM_{10}$ are estimated at 5-km (0.05°) spatial girds across China based on Sentinel-5P and GEOS-FP. In this paper, the Light Gradient Boosting Machine is exploited to train the estimation models, which will fully fuse the multi-source data. For comparison, the Deep Blue AOD product from VIIRS is adopted in a similar framework as a baseline (AOD-based). The validation results show that the ambient concentrations are well estimated through the proposed approach, with the sample-based Cross-Validation $R^2$s and RMSEs of 0.93 (0.9) and 8.982 (17.604) μg/m³ for $PM_{2.5}$ ($PM_{10}$), respectively. Meanwhile, the proposed approach achieves better performance than the AOD-based in different cases (e.g., overall and seasonal). Compared to the related previous works over China, the estimation accuracy of our method is also satisfactory. Furthermore, all the variates of the precursors & chemical compositions for $PM_{2.5}$ and $PM_{10}$ positively contribute to the estimation in the proposed approach, as expected. With regard to the mapping, the estimated results through the proposed approach present consecutive spatial distribution and can exactly express the seasonal variations of $PM_{2.5}$ and $PM_{10}$. It is concluded that the full-coverage estimated results in our study are conducive to the researches on $PM_{2.5}$ and $PM_{10}$ over the regions where the AOD values are missing.





## 1 Introduction


Particulate matters with aerodynamic equivalent diameters less than 2.5 μm (PM$_{2.5}$) and 10 μm (PM$_{10}$) have been
considered as major air pollutants for decades (Finlayson-Pitts et al., 1997; Hall et al., 1992; Lee, 1972), which can hazard
the environment and human health (Crippa et al., 2019; Liu et al., 2020; Ma et al., 2017; Venkataraman et al., 2018). The
ambient concentrations of PM$_{2.5}$ and PM$_{10}$ are strongly relevant to living conditions and required to be accurately
monitored. Generally, ground-based stations are recognized as the most direct and dependable approach to obtain the
ambient concentrations of PM$_{2.5}$ and PM$_{10}$ (Engel-Cox et al., 2013; Li et al., 2017a; Yang et al., 2020a, 2020b).
Nevertheless, the establishing of ground-based stations is costly, which causes difficulties in the implementation (Shen et
al., 2020). Meanwhile, the measurements from ground-based stations are only applicable in small regions and fail to
provide a global perspective (Li et al., 2020). Hence, the approaches based on Chemical Transport Models (CTMs) (Van
Donkelaar et al., 2010; Wang et al., 2016; Weagle et al., 2018) or remote sensing satellites (Chen et al., 2018; Li et al.,
2020; Stafoggia et al., 2019; Shtein et al., 2020; Wei et al., 2019; Yao et al., 2019; You et al., 2015) have been developed
to enlarge the spatial coverage of the PM$_{2.5}$ and PM$_{10}$ monitoring. Since the uncertainties of the emission inventories
adopted in CTMs could be large in some areas (Li et al., 2017b), the approaches based on remote sensing satellites usually
achieve better performance than those based on CTMs.

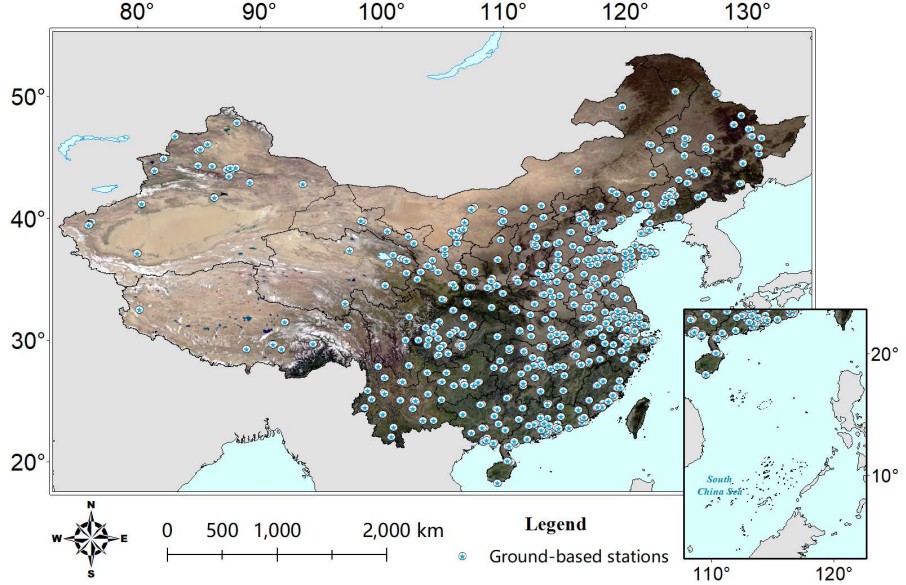


**Figure 1**. The spatial distribution of the ground-based stations over China. The base-map is the true color image of MODIS.
To date, numerous studies have researched on the estimation of the ambient particulate matters concentrations (PM$_{2.5}$ and
PM$_{10}$) using the observations from remote sensing satellites (Chen et al., 2018; Li et al., 2020; Stafoggia et al., 2019;



Shtein et al., 2020; Wei et al., 2019; Yao et al., 2019; You et al., 2015). Thereinto, most of them will adopt a key
atmospheric parameter, i.e., Aerosol Optical Depth (AOD) (Wang et al., 2019a, 2019b), which presents high correlations
with the ambient concentrations of $PM_{2.5}$ and $PM_{10}$ (Guo et al., 2017; Li et al., 2019; Yang et al., 2019). For instance,
Chen et al. (2018) exploited the Random Forest (RF) to acquire the daily ambient concentrations of $PM_{10}$ in China
employing the Deep Blue (DB) and Dark Target (DT) combined AOD products from the Moderate Resolution Imaging
Spectroradiometer (MODIS); Wei et al. (2019) proposed the Space-Time Random Forest model for the mapping of the
daily 1-km ambient concentrations of $PM_{2.5}$ over China on the basis of the Multi-Angle Implementation of Atmospheric
Correction AOD product; Li et al. (2020) developed a brand-new method, i.e., the Geographically and Temporally
Weighted Neural Network, to obtain the daily ambient concentrations of $PM_{2.5}$ across China, which is devised to fix the
spatiotemporal heterogeneous issues of the AOD-$PM_{2.5}$ relationships. There is no doubt that these works have provided
wonderful results and made contributions to the atmospheric environment field. Nevertheless, the data is usually
unavailable in the AOD products from remote sensing satellites due to the influences from clouds, ice/snow, and
arid/semiarid surface (only for DT-like AOD products) (Levy et al., 2013; Sayer et al., 2019). As a consequence, the
completeness of valid values in the estimated results ($PM_{2.5}$ and $PM_{10}$) are also poor through the above-mentioned
approaches, which can result in inconvenience to the researchers. To remedy this deficiency, the algorithm of AOD
recovery is generally utilized as one of the preprocessing steps to fill the missing data in the AOD products. So far, these
algorithms achieve expected performance in local regions (Hua et al., 2019; Xiao et al., 2017) while still likely signify
considerable uncertainties for large scale. Hence, it is necessary to explore a novel approach for the estimation of $PM_{2.5}$
and $PM_{10}$ using other data sources instead of AOD products.
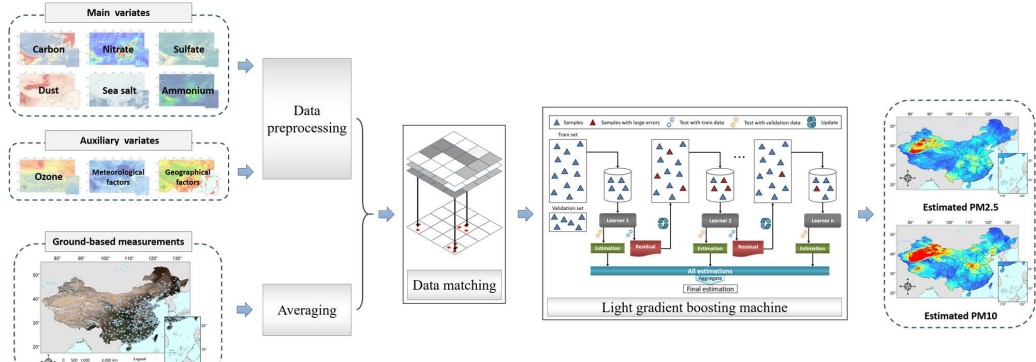

**Figure 2**. The flowchart of the proposed approach in our study. The models for the estimation of $PM_{2.5}$ and $PM_{10}$ are separately trained.
As is well-known, $PM_{2.5}$ and $PM_{10}$ consist of multiple chemical compositions (Dabek-Zlotorzynska et al., 2011; Tao et
al., 2017; Wang et al., 2019c), including sulfate, nitrate, black carbon, dust, etc. In the meantime, some chemical species



are considered as the precursors for $PM_{2.5}$ and $PM_{10}$ (Baker et al., 2007; Heo et al., 2016; Tucker et al., 2000), such as
sulfur dioxide ($SO_2$) and nitrogen dioxide ($NO_2$). It is reasonable to estimate the ambient concentrations of $PM_{2.5}$ and
$PM_{10}$ based on these precursors & chemical compositions. The Sentinel-5 Precursor (Sentinel-5P) satellite (Veefkind et
al., 2012) was launched on 13 October 2017, carrying the TROPOspheric Monitoring Instrument (TROPOMI) to generate
global high-coverage total/tropospheric vertically column of the precursors (e.g., $NO_2$) for $PM_{2.5}$ and $PM_{10}$. Therefore, it
is feasible to adopt the atmospheric products of TROPOMI after the missing data recovery for small regions. However, it
would be insufficient for the estimation of the ambient particulate matters concentrations ($PM_{2.5}$ and $PM_{10}$), only using
the datasets from TROPOMI as the major factors. The GEOS Forward Processing (GEOS-FP) (Lucchesi et al., 2013)
assimilated datasets from the Global Modeling and Assimilation Office (GMAO) can provide the seamless prior
information of the precursors & chemical compositions for $PM_{2.5}$ and $PM_{10}$, which ought to be also introduced as the
major factors in our study.
The purpose of this study is to develop a novel approach to estimate the daily full-coverage 5-km (0.05°) ambient
concentrations of $PM_{2.5}$ and $PM_{10}$ using the datasets from TROPOMI and GEOS-FP. In our study, one of the ensemble
learning methods, i.e., the Light Gradient Boosting Machine (LGBM) (Ke et al., 2017), is applied for the estimation by
fusing the multi-source (TROPOMI, GEOS-FP, and ground-based stations) data. Meanwhile, the DB AOD product from
the Visible Infrared Imager Radiometer Sensor (VIIRS) (Hus et al., 2019) is employed in a similar framework as a baseline
(AOD-based) for comparison, which replaces the atmospheric products of TROPOMI and GEOS-FP. Comprehensive
experiments show that the approach proposed in our study well estimates the ambient particulate matters concentrations
and achieves better performance than the AOD-based, signified in both estimation accuracy and completeness of valid
values.
The remainder of this study is arranged as follows. Section 2 describes the study area and the datasets adopted in our
study. The methodology of the proposed approach is presented in Section 3. Section 4 provides the experiment results,
covering the model performance in different cases (e.g., overall and seasonal), the spatial distribution analyses, and some
discussions. At last, the conclusions are given in Section 5.
**2 Study area and datasets**
**2.1 Study area**
As the country with the largest population in the world (~18% out of the world population by March 2019), China is
regarded as the study area in this paper (shown in Figure 1). For more than ten years, air pollution issues (e.g., high-
polluted particulate matters) are rapidly emerging in China, which results from the acceleration of economic developments



(Wang et al., 2019a). Thanks to the relevant regulations formulated by the government and the endeavors from social
various circles, the air quality has been greatly improved today, including the marked descent of particulate matters (Lin
et al., 2018; Ma et al., 2019). However, the pollutions of particulate matters are not optimistic over China by comparison
with a few developed countries in the world. Meanwhile, $PM_{2.5}$ and $PM_{10}$ are still deemed as the primary air pollutants of
urban areas in the eastern and northwestern China, respectively. It is necessary to develop an approach that can monitor
$PM_{2.5}$ and $PM_{10}$ across China continuously and precisely.

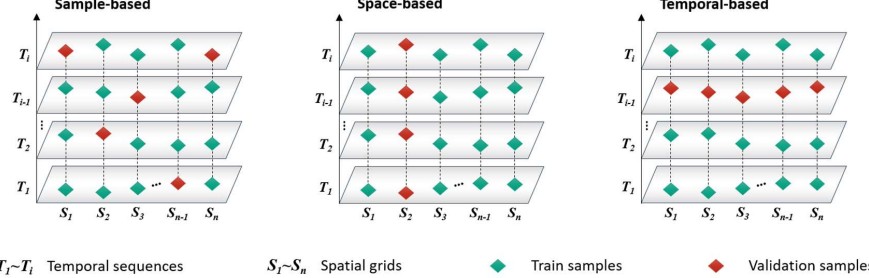

**Figure 3**. The schematic diagram of the validation methods in our study.
**2.2 Datasets**
In this study, the datasets from TROPOMI and GEOS-FP during June 1, 2018 to March 31, 2020 over China are deemed
as the main variates of the inputs in the proposed approach. Meanwhile, some other datasets are adopted as the auxiliary
variates of inputs to enlarge the applicability of the trained models, such as meteorological factors (e.g., planetary
boundary layer height and air temperature), Normalized Difference Vegetation Index (NDVI) (Beck et al., 2006), and
population density (Bai et al., 2018). In addition, the measurements from the China National Environmental Monitoring
Center (CNEMC) are considered as the ground truth-values, consisting of the hourly ambient concentrations of $PM_{2.5}$ and
$PM_{10}$. The descriptions of all datasets are provided as follows.
**2.2.1 Ground-based measurements**
In the study area, the hourly measurements of $PM_{2.5}$ and $PM_{10}$ during June 1, 2018 to March 31, 2020 are firstly allocated
from CNEMC, which can be obtained at http://106.37.208.233:20035/. The spatial distribution of ground-based stations
utilized in this study is demonstrated in Figure 1, using the marks of circles with pentacles inside. As illustrated, a total
of ~1640 ground-based sites (by March 2020) are established in the study area to monitor the pollution of $PM_{2.5}$ and $PM_{10}$,
densely covering most territories of China, except some regions (e.g., Qinghai). The daily ambient concentrations of $PM_{2.5}$
and $PM_{10}$ are deemed as the ground truth-values (output), which are acquired by averaging the hourly measurements
within a day. It's worth noting only the records with no less than 16 hourly measurements in a single day will be adopted.





### 2.2.2 TROPOMI atmospheric products

The TROPOMI is the single instrument of the Sentinel-5P spacecraft (Veefkind et al., 2012), which covers the wavelength
of UltraViolet (UV), Near InfraRed (NIR), and ShortWave InfraRed (SWIR). This hyperspectral spectrometer is devised
to provide daily observations of $SO_2$, $NO_2$, ozone ($O_3$), etc., at high spatial resolutions, using passive remote sensing
methods. The typical pixel size (near-nadir) is set as $7\times3.5$ $km^2$ for all spectral bands, except the UV1 band ($7\times28$ $km^2$)
and SWIR bands ($7\times7$ $km^2$). As for the evaluation, the TROPOMI atmospheric products are routinely compared to ground-
based measurements and observations from other instruments carried onboard remote sensing satellites, such as the Ozone
Monitoring Instrument (Levelt et al., 2006). The evaluation results show that the qualities of the TROPOMI atmospheric
products compile with the mission requirements (Garane et al., 2019; Griffin et al., 2019; Theys et al., 2017). In our study,
the records of "sulfurdioxide_total_vertical_column_1km" and "nitrogendioxide_tropospheric_column" are regarded as
the main variates in the proposed approach, which are related to sulfate and nitrate, respectively. In addition, particulate
matters ($PM_{2.5}$ and $PM_{10}$) were discovered to be associated with $O_3$ (Chen et al., 2019, 2020). Therefore, the record of
"ozone_total_vertical_column" is also introduced in the proposed approach as one of the auxiliary variates. The
information about the TROPOMI atmospheric products used in this study is specifically provided in Table S1 of the
supplementary materials.

### 2.2.3 GEOS-FP assimilated products

The GEOS-FP data assimilation system employs an analysis designed collectively with the National Centers for
Environmental Prediction (Lucchesi et al., 2013), which is the current operational met data product from GMAO.
Generally, the GEOS-FP can provide the time-averaged (e.g., hourly) assimilated datasets performed at a spatial resolution
of $0.25º \times 0.3125º$, including the atmospheric chemical species and meteorological factors. In our study, the records of the
precursor/chemical compositions for $PM_{2.5}$ and $PM_{10}$ from GEOS-FP are considered as the main variates of the inputs,
including the nitrate-related (i.e., Nitrate Column Mass Concentration), carbon-related (e.g., Organic Carbon Column
Mass Concentration), sulfate-related (i.e., SO4 Column Mass Density), etc. Furthermore, a few meteorological factors
from GEOS-FP are also adopted as the auxiliary variates in the proposed approach, such as wind speed, specific humidity,
and planetary boundary layer height. The relevant information of the GEOS-FP datasets used in our study is presented in
the supplementary materials (see Table S1).

### 2.2.4 Geographical factors

Some geographical factors are usually exploited as the ancillary variates to estimate the ambient concentrations of $PM_{2.5}$
and $PM_{10}$ in previous studies, including the land cover classifications (Zhang et al., 2017), population density, NDVI, and





road density (Haklay et al., 2008). Hence, these factors are also introduced in our study, which are associated with $PM_{2.5}$
and $PM_{10}$. The detailed information about the geographical factors utilized in our study is listed in Table S1 of
supplementary materials, which will not be repeatedly described here.

**2.2.5 VIIRS DB AOD product**

The DB algorithm (Hsu et al., 2019) was first proposed to retrieve aerosol properties of the observations from MODIS
over arid/semiarid and urban areas. After a decade, an enhanced DB algorithm was developed and applicable for all areas
without snow/ice. In the latest Collection 6.1 (C6.1), the scheme of DB was upgraded once again with several updates,
such as the heavy smoke detection. With regard to VIIRS, the procedures are similar to the one for MODIS in C6.1, while
a few marked differences still exist. For example, a modified NIR method is employed to acquire the surface reflectance
in croplands. The evaluation results showed that the VIIRS DB algorithm performs better than the one for MODIS over
Asia (Wang et al., 2020). Due to the similar spatial resolution (6-km) with TROPOMI, the DB AOD from VIIRS is
deemed as the main variate in a framework (baseline, AOD-based) for comparison, which is close to the proposed
approach (with the same auxiliary variates expect the $O_3$ product from TROPOMI). The specific information about the
VIIRS DB AOD product is appended in the supplementary materials (Table S2).

**3 Methodology**

The flowchart of the proposed approach is depicted in Figure 2. As can be seen, the datasets (main and auxiliary variates)
are initially preprocessed in advance of being adopted as the inputs, e.g., the resampling and missing data recovery.
Meanwhile, the ground truth-values (output) are obtained by averaging the hourly ground-based measurements within a
day ($\geq$ 16 out of 24). Next, the inputs and ground truth-values ought to be spatially matched considering the differences
between them. After the data matching, the data pairs (matched samples) will be fed into the LGBM to train the model.
Eventually, a total of three 10-fold Cross-Validation (CV) methods are exploited to validate the performance of the
proposed approach. The specific procedures are stated in the following subsections. It's worth noting that the models for
the estimation of $PM_{2.5}$ and $PM_{10}$ are separately trained. In addition, the methodology of the baseline (AOD-based) is
close to the proposed approach, which is appended in Figure S1 of the supplementary materials.

**3.1 Data preprocessing**

Firstly, the spatial resolutions of the datasets (main and auxiliary variates) should be adjusted to coincident. In our study,
the datasets from TROPOMI, GEOS-FP, and geographical factors are resampled to 5-km through the nearest neighbor
interpolation (Olivier et al., 2012), bicubic interpolation (Nuno-Maganda et al., 2005), and area-weighted aggregation
(Liu et al., 2019), respectively. In the meantime, the daily datasets of GEOS-FP are acquired by averaging the hourly/3-
hour records within a day. Next, the missing values for small regions in the datasets from TROPOMI are filled through
the exemplar-based algorithm (Criminisi et al., 2004). Since the missing coverage of the TROPOMI $SO_2$ and $O_3$ products
is little, only the examples of the simulated experiments for the TROPOMI $NO_2$ product are demonstrated in the
supplementary materials (Figure S2). Besides, the missing values for some pixels in the NDVI product are also filled
using the Inverse Distance Weighted interpolation (Wang et al., 2019b).

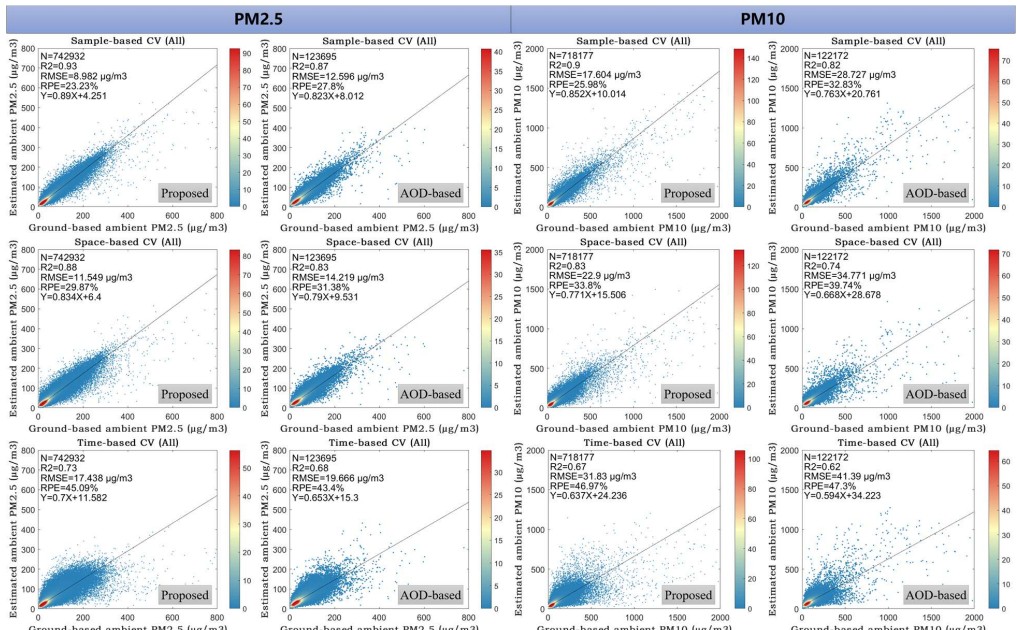

**Figure 4**. The density scatter plots of the validation results in the study area. The black solid line signifies the fitted line and the color
bar denotes the density of samples. Y: estimated ambient concentrations of $PM_{2.5}$ and $PM_{10}$; X: ground-based ambient concentrations
of $PM_{2.5}$ and $PM_{10}$.
**3.2 Data matching**
Generally, the datasets (main and auxiliary variates) are grid-based at different spatial resolutions, while ground-based
stations only measure the ambient concentrations of $PM_{2.5}$ and $PM_{10}$ for small regions. Therefore, the grid-based datasets
and ground-based measurements should be spatially matched. In brief, all the ground truth-values falling in one spatial
grid (5-km) are averaged to match the datasets from TROPOMI, GEOS-FP, and geographical factors.
**3.2 Light Gradient Boosting Machine**
LGBM is a newly devised and advanced ensemble learning method based on the Gradient Boosting Decision Tree (Ke et
al., 2017). As one of the gradient boosting algorithms, the targets for each training round in LGBM are residual, which



are computed from the truth-value and the estimations after previous training rounds. In other words, the learners in
LGBM are mutually associated and consequently the dependencies between learners will be employed. For instance, the
overall performance can be significantly improved by assigning higher weights to the samples estimated with larger errors
in previous training rounds. Compared to previous gradient boosting algorithms, LGBM is capable of easily achieving
higher accuracy with fewer sample features, less memory, and faster speed. In general, the highlights of LGBM mainly
consists of two parts: Gradient-based One-Side Sampling and Exclusive Feature Bundling. Both of them are designed to
decrease the number of samples in each training round and retained the estimation accuracy. The specific structures of
LGBM are complicated and will not be described in our study. For more information, readers could refer to Ke et al.,

209  2017.

LGBM can process high-dimensional big data of large scale, presenting higher efficiency and better performance by
comparison with conventional machine learning methods, e.g., the RF, Generalized Regression Neural Network
(Cigizoglu et al., 2005), and Support Vector Regression (Drucker et al., 1997). Hence, it is reasonable to adopt LGBM in
our study. The general scheme of the model for estimating the ambient concentrations of $PM_{2.5}$ and $PM_{10}$ can be expressed
as Eq. (1).
$C_{PM} = f(VM_P, VM_{CC}, VA_{O3}, VA_{MF}, VA_{GF})$ (1)
where $C_{PM}$ signifies the estimated ambient concentrations of $PM_{2.5}$ and $PM_{10}$. $f$ denotes the estimation function for the
ambient concentrations of $PM_{2.5}$ and $PM_{10}$ based on LGBM. $VM_P$ and $VM_{CC}$ include the main variates of the precursors
and chemical compositions, respectively, for $PM_{2.5}$ and $PM_{10}$. $VA_{O3}$, $VA_{MF}$, and $VA_{GF}$ represent the auxiliary variates of
the $O_3$ from TROPOMI, meteorological factors, and geographical factors, respectively. The detailed information about
each variate can be found in Table S1 and S3 of the supplementary materials. The setting of the LGBM parameters is
listed in Table S4.
**3.3 Validation methods**
To sufficiently validate the performance of the proposed approach, a total of three 10-fold CV methods, i.e., the sample-
based CV, space-based CV, and time-based CV, are exploited in our study. With regard to the sample-based CV, all the
matched samples are divided into 10 folds at random (the number is approximately identical). Next, nine folds are
employed to train the model and the remaining one is considered for the validation. At last, the previous step is repeatedly
performed 10 times and consequently each fold can be validated. As for the space-based CV and time-based CV, the steps
are close to those for the sample-based CV. The only distinction is that the 5-km spatial grids (space-based CV) or temporal
sequences (time-based CV) are randomly separated into 10 folds, rather than the matched samples. The schematic diagram





of the three 10-fold CV methods is illustrated in Figure 3. In this study, the estimated results are validated through three
metrics: the coefficient of determination ($R^2$), the Root Mean Square Error (RMSE), and the Relative Percentage Error
(RPE). It is worth noting that all the metrics are computed at the significance levels of $p < 0.01$ in our study.
**Table 1**. The validation results for the proposed and AOD-based considering whether the values of VIIRS DB AOD are missing. VR:
valid regions (the values of VIIRS DB AOD are available); MR: missing regions (the values of VIIRS DB AOD are unavailable); T:
true; F: false.

| CV method | Region | Approach | PM$_{2.5}$ | | | | PM$_{10}$ | | | |
|---|---|---|---|---|---|---|---|---|---|---|
| | | | N | $R^2$ | RMSE | RPE | N | $R^2$ | RMSE | RPE |
| Sample-based | VR | Proposed | 122614 | 0.92 | 9.753 μg/m³ | 21.61% | 121098 | 0.89 | 22.295 μg/m³ | 25.53% |
| | | AOD-based | | 0.87 | 12.535 μg/m³ | 27.77% | | 0.82 | 28.436 μg/m³ | 32.57% |
| | MR | Proposed | 620742 | 0.93 | 8.826 μg/m³ | 23.61% | 597471 | 0.9 | 16.517 μg/m³ | 25.9% |
| Space-based | VR | Proposed | 122614 | 0.87 | 12.43 μg/m³ | 27.54% | 121098 | 0.82 | 28.878 μg/m³ | 33.07% |
| | | AOD-based | | 0.83 | 14.311 μg/m³ | 31.7% | | 0.74 | 34.803 μg/m³ | 39.86% |
| | MR | Proposed | 620742 | 0.88 | 11.691 μg/m³ | 31.28% | 597471 | 0.83 | 21.629 μg/m³ | 33.92% |
| Time-based | VR | Proposed | 122614 | 0.71 | 18.795 μg/m³ | 41.64% | 121098 | 0.65 | 39.906 μg/m³ | 45.7% |
| | | AOD-based | | 0.68 | 19.58 μg/m³ | 43.38% | | 0.62 | 41.181 μg/m³ | 47.16% |
| | MR | Proposed | 620742 | 0.73 | 17.153 μg/m³ | 45.89% | 597471 | 0.67 | 29.91 μg/m³ | 46.91% |

Note: The numbers of the matched samples in VR are less than those for the AOD-based (see Figure 4) since the original swath files
of TROPOMI are not available on several days.

## 4 Experiment results and discussions

### 4.1 Overall validation results

The density scatter plots of the sample-based CV, space-based CV, and time-based CV for the estimated ambient
concentrations of PM$_{2.5}$ and PM$_{10}$ are depicted in Figure 4. As can be seen, the estimated concentrations through the
proposed approach are validated with sufficient matched samples (742932 and 718177) in the study area, indicating the
reliability of the validation results. By contrast, the number of matched samples for the AOD-based (123695 and 122172)
is much less due to the missing values in the VIIRS DB AOD product. As for all matched samples, the estimated ambient
concentrations of PM$_{2.5}$ and PM$_{10}$ through the proposed approach achieve a better performance compared to those through
the AOD-based, with higher $R^2$s for three CV methods (e.g., PM$_{2.5}$: 0.93, 0.88, and 0.73). In the meantime, the
performance difference of the estimation between PM$_{2.5}$ and PM$_{10}$ for the proposed approach is smaller than that for the
AOD-based, suggesting the robustness and applicability of our approach. To further validate the proposed approach, the



experiment results of some related previous works over China are provided in the supplementary materials. It is worth
noting that only the metrics computed from the estimated results of 2019 (a whole year) in our study are presented for
fairness. As listed in Table S5, the proposed approach shows a satisfactory performance by comparison with these works,
which is reflected in the estimation accuracy or completeness of valid values.

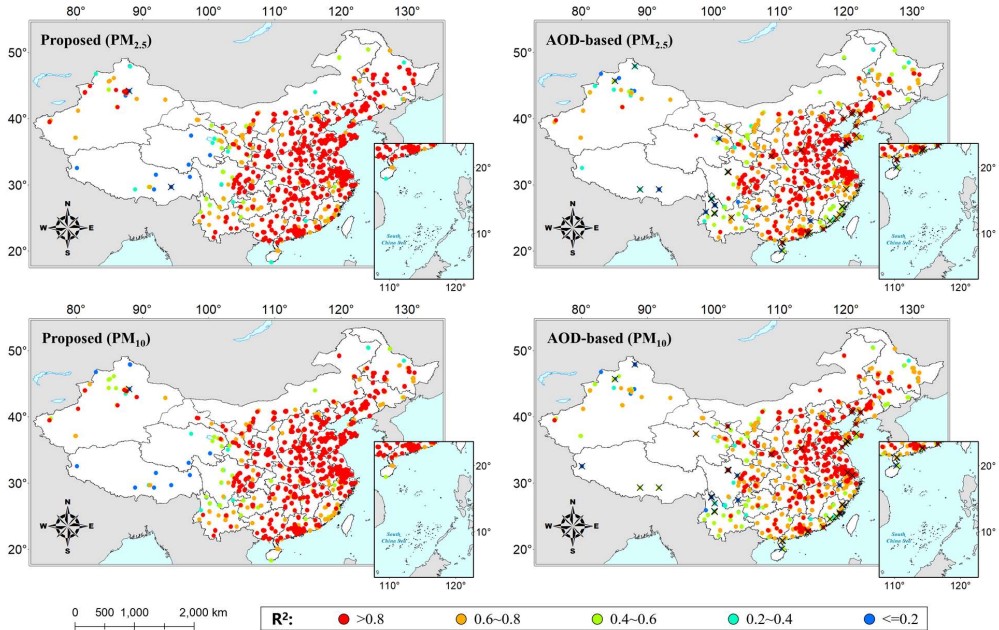


**Figure 5**. The spatial distribution of $R^2$s for the space-based CV at each matched grid over China. The black crosses denote that the
significance levels (p) of the metrics are not less than 0.01 at these matched grids.
**4.2 Seasonal and regional validation results**
The density scatter plots of three CV methods for four seasons (2019), i.e., DJF (Dec., Jan., and Feb.), MAM (Mar., Apr.,
and May.), JJA (Jun., Jul., and Aug.), and SON (Sep., Oct., and Nov.), are appended in the supplementary materials. As
demonstrated in Figure S3-S6, the performance of the proposed approach is also as expected in different seasons, of which
the metrics generally overmatch those of the AOD-based, especially for JJA. Next, the matched samples are divided into
two parts according to whether the values of VIIRS DB AOD are missing to compare the proposed approach and the
AOD-based under the equal condition. As listed in Table 1, the proposed approach presents a superior estimation accuracy
of $PM_{2.5}$ and $PM_{10}$ for three CV methods in the valid regions, with differences of 0.03-0.08 in $R^2$s and 1.46-7.04% in
RPEs. Besides, it's observed that the proposed approach performs well in the missing regions, showing similar metrics
to those in the valid regions.

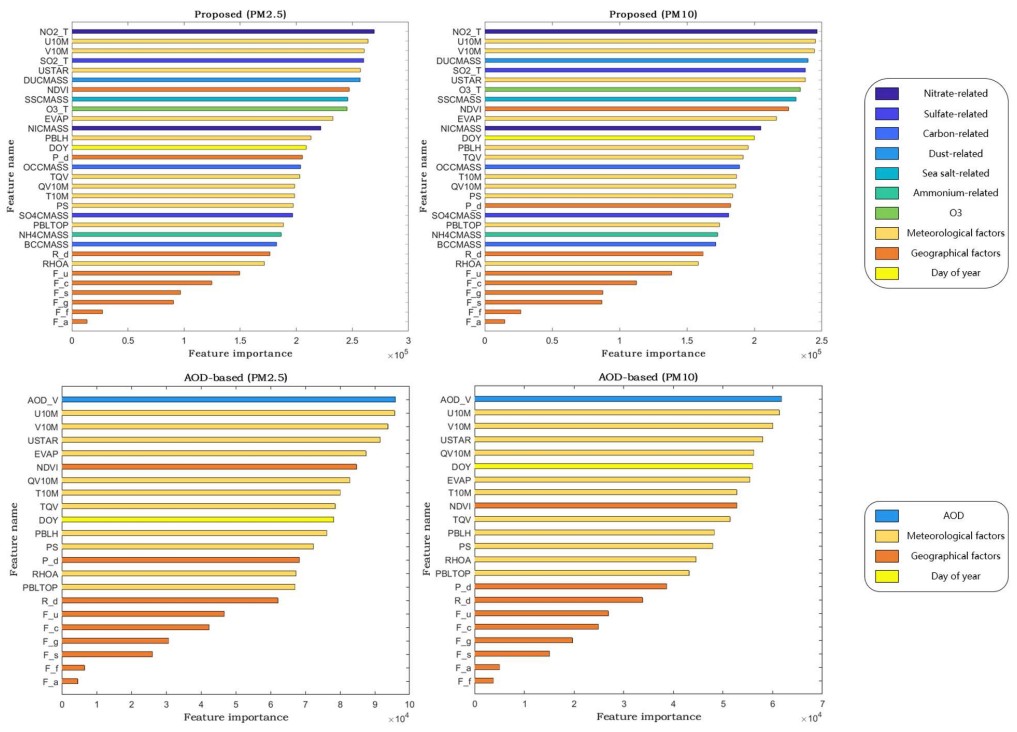

**Figure 6**. The bar graphs of the feature importance for the proposed and AOD-based. The full names of the features can be found in Table S3.

## 4.3 Grid-based validation results

The performance at each matched grid is important, which is able to reveal the influence from the spatial heterogeneity of PM$_{2.5}$ and PM$_{10}$. Since the division of matched samples as per spatial grids could represent the spatial patterns, the space-based CV results are adopted to map the spatial distributions of the metrics at each matched grid in our study. As shown in Figure 5, a total of 974/79.6% and 945/77.27% matched grids present the R$^2$s>0.8 (p<0.01) of PM$_{2.5}$ and PM$_{10}$ for the proposed approach, respectively. In contrast, the numbers of the matched grids showing the R$^2$s>0.8 (p<0.01) visibly reduce (by 352/28.78% of PM$_{2.5}$ and 420/34.34% of PM$_{10}$) for the AOD-based. Meanwhile, the proposed approach also displays higher R$^2$s compared to the AOD-based in some regions, where the ground-based stations are sparse, such as Xinjiang. In addition, the spatial distributions of RMSEs, RPEs, and sample numbers for the space-based CV at each matched grid are appended in Figure S7-S9 of the supplementary materials. Since RMSE is one of the absolute metrics, which are relevant to the magnitudes, the spatial distribution distinctions of RMSEs at matched grids for the proposed and AOD-based will be not discussed. By comparison with R$^2$s, the differences of the matched grids between the proposed and AOD-based are smaller for RPEs (<=30%, p<0.01), with the numbers of 126/10.3% and 150/12.26% of PM$_{2.5}$ and


PM$_{10}$, respectively. From Figure S9, most of the matched grids exceed 600 samples for the proposed approach, while
almost all the sample numbers of the AOD-based are less than 300. As a consequence, the non-significant metrics
(p>=0.01) are numerous in the space-based CV results through the AOD-based due to the missing coverage.

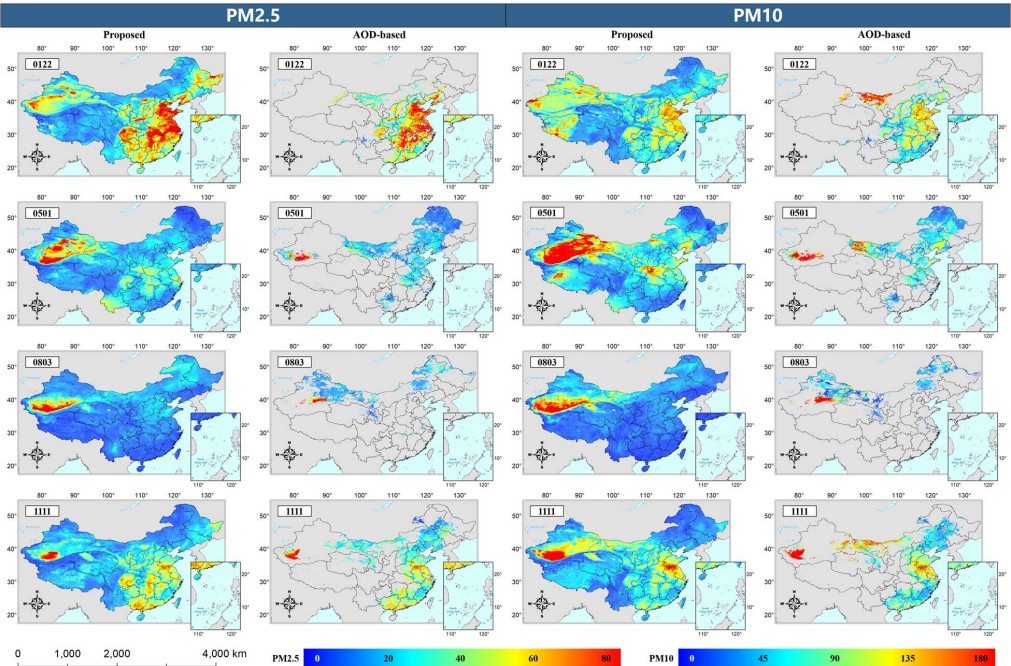


**Figure 7**. The daily estimated ambient concentrations of PM$_{2.5}$ and PM$_{10}$ for the proposed and AOD-based across China in 2019. The
color bars represent the values of the estimated results. Units: μg/m$^3$.

## 4.4 Feature importance of variates

The bar (pie) graphs that provide the feature importance (percentages) of the inputs in the proposed and AOD-based are
illustrated in Figure 6. With regard to the proposed approach, the variates from TROPOMI, i.e., NO2_T and SO2_T, play
an important part in estimating the results, which are the precursors for PM$_{2.5}$ and PM$_{10}$. In the meantime, the rank of
DUCMASS rises for the estimation of PM$_{10}$ compared to that of PM$_{2.5}$, indicating the flexibility of our approach.
Furthermore, all the variates of the precursors & chemical compositions for PM$_{2.5}$ and PM$_{10}$ (e.g., carbon-related)
positively contribute to the estimation through the proposed approach, which is as expected. By contrast, most of the
contributions in the results estimated by the AOD-based mainly stem from the meteorological factors.

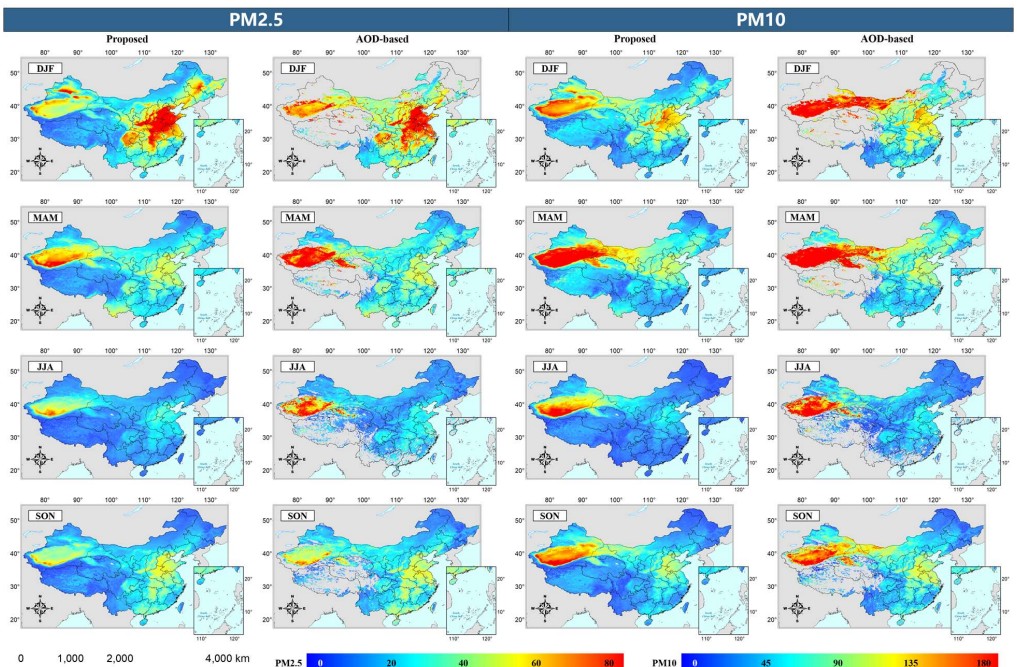


**Figure 8**. The seasonal estimated ambient concentrations of PM$_{2.5}$ and PM$_{10}$ for the proposed and AOD-based across China in 2019.
The color bars represent the values of the estimated results. Units: μg/m$^3$.

## 4.5 Evaluation of the spatial distribution

At first, the estimated ambient concentrations of PM$_{2.5}$ and PM$_{10}$ for a total of four days, i.e., 20190122, 20190501, 20190803, and 20191111, are utilized to evaluate the spatial distribution of the daily estimated results. As demonstrated in Figure 7, the daily estimated results through the proposed approach present consecutive spatial distribution without distinctly incorrect structures, suggesting that our approach is reliable. On the contrary, the absence of a large scale can be discovered in the daily ambient concentrations of PM$_{2.5}$ and PM$_{10}$ estimated by the AOD-based. Next, the estimated results for four seasons in 2019 are also mapped to evaluate the seasonal spatial distribution. As illustrated in Figure 8, the proposed approach is capable of exactly expressing the seasonal variations of PM$_{2.5}$ and PM$_{10}$. For instance, the high values of the seasonal estimated PM$_{2.5}$ principally emerge in DJF, which is caused by the heating emissions (e.g., fossil fuels combustion) and adverse meteorological conditions (Cao et al., 2012); The seasonal estimated ambient concentrations of PM$_{10}$ mainly appear large in MAM due to the sand storms and dry weathers (Li et al., 2017c). With regard to most areas of China (except the Northwest), the seasonal estimated results through the proposed and AOD-based display similar spatial patterns, with the distinctions of the values. In DJF, the differences between the proposed and AOD-based are the greatest for four seasons, which likely results from the influence of the missing values (AOD) on time-averaged results. Meanwhile, it is observed that the seasonal estimated ambient concentrations of PM$_{2.5}$ and PM$_{10}$ through

the proposed are generally larger than those through the AOD-based in arid/semiarid regions, such as Xinjiang. As stated
in Section 4.3, the proposed approach shows higher $R^2$s at matched grids compared to the AOD-based in Xinjiang.
Therefore, the discrepancy possibly derives from the overestimation of VIIRS DB AOD in arid/semiarid regions (Sayer
et al., 2019; Wang et al., 2020).

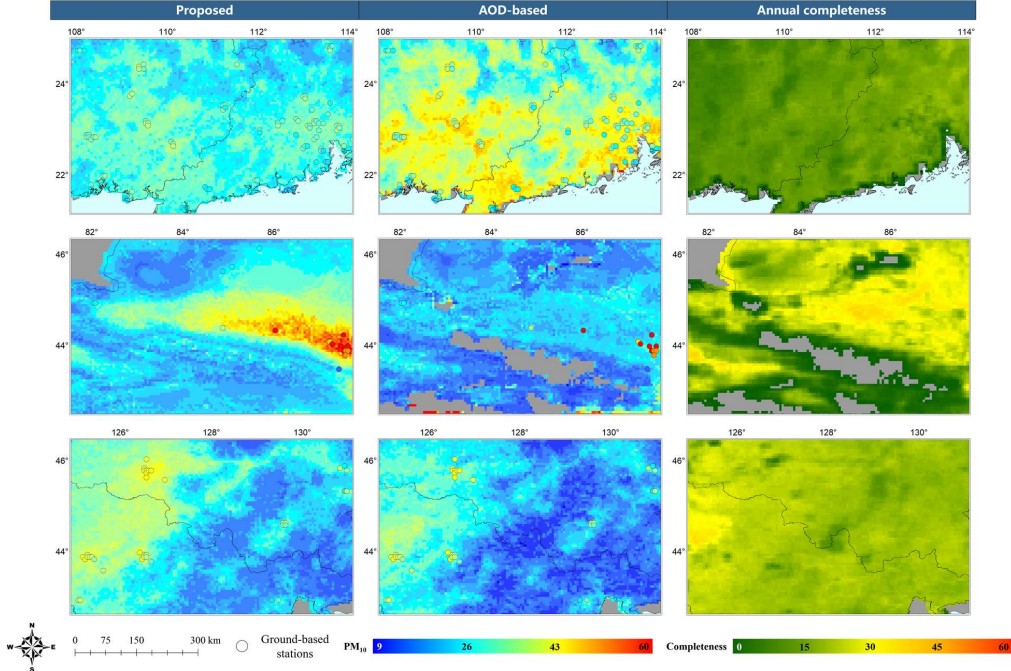

**Figure 9**. The annual estimated ambient concentrations of PM$_{2.5}$ for the proposed and AOD-based over local regions in 2019. The left
color bar represents the values of the estimated results and ground truth-values. The right color bar denotes the completeness of VIIRS
DB AOD. Units: μg/m$^3$ for PM$_{2.5}$ and % for completeness.

**4.6 Discussions of the time-averaged results**

To further explore the influence of the missing values (AOD) on time-averaged results, the annual estimated ambient
concentrations of PM$_{2.5}$ and PM$_{10}$ are mapped over local regions in Figure 9 and S10. In the meantime, the annual ground
truth-values are also provided in the figures, which is conducive to indicating the real spatial distribution of PM$_{2.5}$ and
PM$_{10}$. As depicted in Figure 9, the annual estimated ambient concentrations of PM$_{2.5}$ through the AOD-based present
great distinctions by comparison with the ground truth-values in the selected regions. This suggests that the influence of
the missing values in the AOD product on time-averaged results is nonnegligible. Namely, the AOD-based likely
incorrectly estimates the time-averaged (e.g., annual) ambient concentrations of PM$_{2.5}$ in some regions. By contrast, the
proposed approach achieves a satisfactory performance compared to the ground truth-values. As for PM$_{10}$, the discovery





is similar (see Figure S10) and will not be repeatedly stated. The full-coverage results estimated by the proposed approach
are conducive to the researches on PM$_{2.5}$ and PM$_{10}$ over the regions where the AOD values are missing. In addition, the
box plots displaying the variations of the absolute relative difference between annual estimated results (over China)
through the proposed and AOD-based (see Eq. s1) with the increment of annual AOD completeness are shown in Figure
10. It can be observed that the overall means of the absolute relative difference are 26.54% and 29.78% for PM$_{2.5}$ and
PM$_{10}$, respectively. Meanwhile, the absolute relative difference (mean) and annual AOD completeness appear negative
correlations, especially for the regions where the AOD values are largely missing (<20%).

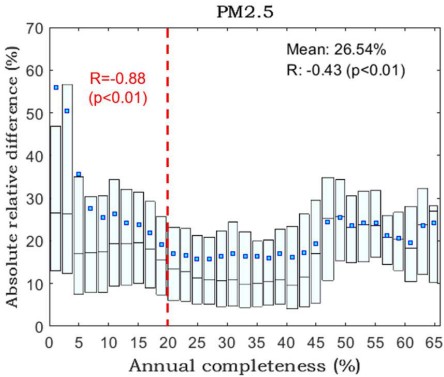
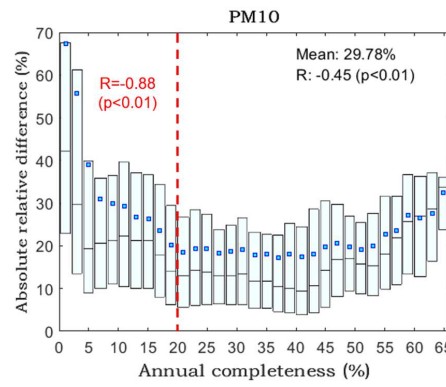


**Figure 10**. The variations (box plots) of the absolute relative difference between annual estimated results (over China) through the
proposed and AOD-based with the increment of annual AOD completeness. For each box, the middle line, rectangle dot, top, and
bottom hinges are the median, mean, 25$^{th}$, and 75$^{th}$ percentiles, respectively.
**5 Conclusions**
In this study, a novel approach is developed, which can estimate the daily full-coverage ambient concentrations of PM$_{2.5}$
and PM$_{10}$ considering their precursors & chemical compositions at a 5-km (0.05°) spatial resolution over China from
TROPOMI and GEOS-FP. To sufficiently fuse the multi-source data, one of the ensemble learning methods, i.e., LGBM,
is employed to train the estimation models. In the meantime, the DB AOD product from VIIRS is applied in a similar
framework (AOD-based) for comparison. The validation results show that the ambient concentrations are well estimated
through the proposed approach in the study area, with the sample-based CV R$^2$s and RMSEs of 0.93 (0.9) and 8.982
(17.604) μg/m$^3$ for PM$_{2.5}$ (PM$_{10}$), respectively. Meanwhile, the proposed approach achieves better performance than the
AOD-based in different situations (e.g., overall and seasonal), suggesting that our approach is reliable. Compared to the
related previous works, the estimation accuracy of the proposed approach is also satisfactory. For the feature importance,
all the variates of the precursors & chemical compositions for PM$_{2.5}$ and PM$_{10}$ (e.g., carbon-related) positively contribute
to the estimation in our approach, which is as expected. As for the mapping, the estimated results through the proposed



approach appear consecutive spatial distribution without visibly incorrect structures and can exactly express the seasonal
variations of PM$_{2.5}$ and PM$_{10}$. In addition, it is discovered that the AOD-based likely incorrectly estimates the time-
averaged ambient concentrations of PM$_{2.5}$ and PM$_{10}$. The full-coverage estimated results through the proposed approach
are conducive to the studies on PM$_{2.5}$ and PM$_{10}$ in the regions where the AOD values are missing.

**Author contributions**

YW designed the study, collected and processed the data, analyzed the results, and wrote the paper. QQY provided
constructive comments on the paper. TWL, SYT, and LPZ revised the paper. All authors contributed to the study.

**Acknowledgments**

This work was supported by the National Natural Science Foundation of China (No. 41922008). The authors would like
to be greatly grateful to the institutions for providing the datasets used in this paper.

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
