# Peer review of "Estimating daily full-coverage and high-accuracy 5-km ambient"

_Atmospheric Chemistry and Physics, 2020_

## Short Comment (SC1) · 11 Oct 2020

This study presents an admirable, solid work of estimating PM concentrations using data sources other than conventional AOD products. The manuscript is clearly well-written and easy to follow. From my humble view, I am only concerned with the high-resolution use of the coarse-resolution GEOS-FP datasets (further explained below). If this can be properly justified, I believe it will be a very nice paper.

The authors explained that it is reasonable to estimate PM concentrations using
datasets of chemical precursors and species. But because the high-resolution TROPOMI only provides chemical precursors, the authors employed the coarse-resolution GEOS-FP for chemical species. However, using coarse-resolution datasets as MAJOR predictors (also confirmed by their relatively high ranks shown in Figure 6) for high-resolution PM mapping inevitably introduced uncertainties rather than more valuable information. This is simply because we typically do not have accurate, high-resolution emission inventories to drive a data assimilation system like GEOS-FP. The authors should justify this issue. Otherwise it would be possible to doubt the significance of this study. In other words, because GEOS-FP provides PM species data and thereby can provide total PM data at a coarse resolution via some sort of add-up, is it really necessary to do a big load of modeling (correlating) work to derive PM concentrations with a plausible high resolution but associated inevitable uncertainties?

Other minor issues:

The GEOS-FP provides more than described variables for use. Why do you choose the column mass density variable rather than others, say surface concentration variables?

Line 279-280, because you are modeling on the same dependent variable (i.e. PM concentrations), RMSE is comparable though you can choose not to describe for conciseness.

Line 314, larger => smaller?

**ACPD**

---

## Referee Comment (RC1) · Anonymous Referee #1 · 24 Oct 2020

**Review of Wang et al 2020**

Wang et al. present an analysis of particulate matter over China. They use a Light Gradient Boosting Machine regression technique to combine satellite observations and model simulations to estimate surface $PM_{2.5}$ and $PM_{10}$. While the broad topics addressed in this manuscript (i.e. pollutant estimation, data-model fusion, machine learning, etc.) are important areas of research, I cannot recommend this paper for publication due to serious methodological issues and the fact that very similar and more comprehensive work has already been published.

**Similarity to previous work**

Van Donkelaar et al. (2019) and Hammer et al. (2020) use a far simpler statistical method to achieve similar performance at across larger temporal, spatial, and chemical scales. As a specific example, Hammer et al. (2020) use a relatively simple linear adjustment of satellite observations to achieve similar performance at similar resolution globally for 20 years.

**Methodological Issues**

Section 3.1

This data processing is inappropriate for the input data as described in the supplement. In particular, downscaling wind fields through bicubic interpolation does not preserve mass or energy, nor does it accurately reproduce any higher order variability in the wind fields. More advanced statistical approaches (e.g. Kirchmeier et al., 2014) are necessary to produce physically consistent information.

Section 3.3

The 10x cross validation scheme described in this section does not account for the very large amount of spatial and temporal correlation present in environmental data. This will substantially and inappropriately bias the estimated skill of the machine learning model (e.g. Hastie et al. 2001, Roberts et al., 2017). The authors should perform a more rigorous evaluation of the model skill consistent with the standard in the field (e.g. Barnes et al., 2020). For example, the authors could cross validate through block methods by removing longer time periods of data or removing entire spatial regions of data beyond the autocorrelation length scale.

Section 4.4.

The feature importance results in this section call in question the validity of the modelling approach here. The top ranked variables include those that were unphysically downscaled: wind speeds (U, V, and Ustar) and turbulent evaporation, as well as total column ozone. As the vast majority of the ozone within a total column is present in the stratosphere, any surface predictive capacity associated with that variable is dubious.

Additionally, the high importance ranking of $NO_2$ potentially indicates that the machine learning model is predicting based simply off the proximity to combustion sources. This explains why the model performance is so poor over more remote regions in Western China and limits the applicability of the method developed.

Figures

All figures with maps in this paper violate the ACP maps policy:
https://www.atmospheric-chemistry-and-physics.net/submission.html#mapsaerials
"In order to depoliticize scientific articles, authors should avoid the drawing of borders or use of contested topographical names."
The inset of the South China Sea does not aid in the scientific interpretation of the results presented in this manuscript in any way, and only confuses the figures.

**Specific Comments:**
L14 "Most of the existing approaches for the estimation of $PM_{2.5}$ and $PM_{10}$ employed the remote sensing Aerosol Optical Depth (AOD) products as the main variate."

I don't believe this is the case. Most approaches to estimate $PM_{2.5}$ and $PM_{10}$ come from in situ observations of aerosol mass and size distributions.

L29 "conducive to the researches". I'm not sure what this sentence means.

L38 Satellite observations are much more expensive than ground-based monitoring.

L45 This statement needs appropriate citation.

Section 2.2
This section is far too lacking in details to interpret or reproduce the work presented in this paper. The authors should explicitly state the variables used in the main body of the manuscript.

L120 This link does not work for me.

Section 3.2
How were the hyperparameters selected? Was there any optimization or search algorithm applied here?

Figure 5. The colors being split into only 5 bins makes assessing performance difficult. Consider using a continuous colorbar.

L329 The authors should explicitly show evidence for this incorrect estimation through sampling.

**References**
Barnes, Elizabeth A., Benjamin Toms, James W. Hurrell, Imme Ebert-Uphoff, Chuck Anderson, and David Anderson. "Indicator Patterns of Forced Change Learned by an Artificial Neural Network." Journal of Advances in Modeling Earth Systems 12, no. 9 (2020): e2020MS002195. https://doi.org/10.1029/2020MS002195.

Barnes, Elizabeth A., James W. Hurrell, Imme Ebert-Uphoff, Chuck Anderson, and David Anderson. "Viewing Forced Climate Patterns through an AI Lens." Geophysical Research Letters 2019. https://doi.org/10.1029/2019GL084944.

Hammer, Melanie S., Aaron van Donkelaar, Chi Li, Alexei Lyapustin, Andrew M. Sayer, N. Christina Hsu, Robert C. Levy, et al. "Global Estimates and Long-Term Trends of Fine Particulate Matter Concentrations (1998–2018)." Environmental Science & Technology 54, no. 13 (2020): 7879–90. https://doi.org/10.1021/acs.est.0c01764.
Hastie, Trevor, Robert Tibshirani, and Jerome Friedman. The Elements of Statistical Learning. Springer Series in Statistics. New York, NY, USA: Springer New York Inc., 2001.

Kirchmeier, Megan C., David J. Lorenz, and Daniel J. Vimont. "Statistical Downscaling of Daily Wind Speed Variations." Journal of Applied Meteorology and Climatology 53, no. 3 (2014): 660–75. https://doi.org/10.1175/JAMC-D-13-0230.1.

Roberts, David R., Volker Bahn, Simone Ciuti, Mark S. Boyce, Jane Elith, Gurutzeta Guillera-Arroita, Severin Hauenstein, et al. "Cross-Validation Strategies for Data with Temporal, Spatial, Hierarchical, or Phylogenetic Structure." Ecography 40, no. 8 (2017): 913–29. https://doi.org/10.1111/ecog.02881.

Van Donkelaar, A, Randall V. Martin, Chi Li, and Richard T. Burnett. "Regional Estimates of Chemical Composition of Fine Particulate Matter Using a Combined Geoscience-Statistical Method with Information from Satellites, Models, and Monitors." Environmental Science & Technology 2019. https://doi.org/10.1021/acs.est.8b06392.

---

## Author Comment (AC1) · 12 Nov 2020

**Response to Comments on the Manuscript (acp-2020-1004):**

**"Estimating daily full-coverage and high-accuracy 5-km ambient particulate matters across China: considering their precursors and chemical compositions"**
* * *
**Response to Comments of Fei Yao:**

**General comment:**

This study presents an admirable, solid work of estimating PM concentrations using data sources other than conventional AOD products. The manuscript is clearly well-written and easy to follow. From my humble view, I am only concerned with the high-resolution use of the coarse-resolution GEOS-FP datasets (further explained below). If this can be properly justified, I believe it will be a very nice paper.

**Response:** We are particularly grateful for your approval of our research. An item-by-item response to the constructive comments follows. Thanks for your time.

**Major comment:**

**Q1:** The authors explained that it is reasonable to estimate PM concentrations using datasets of chemical precursors and species. But because the high-resolution TROPOMI only provides chemical precursors, the authors employed the coarse-resolution GEOS-FP for chemical species. However, using coarse-resolution datasets as MAJOR predictors (also confirmed by their relatively high ranks shown in Figure 6) for high-resolution PM mapping inevitably introduced uncertainties rather than more valuable information. This is simply because we typically do not have accurate, high-resolution emission inventories to drive a data assimilation system like GEOS-FP. The authors should justify this issue. Otherwise it would be possible to doubt the significance of this study. In other words, because GEOS-FP provides PM species data and thereby can provide total PM data at a coarse resolution via some sort of add-up, is it really

**necessary to do a big load of modeling (correlating) work to derive PM concentrations with a plausible high resolution but associated inevitable uncertainties?**

**Response:** Thank you for your significant comment. According to the previous works [1-2] of our team, the high-resolution geographical factors (e.g., land cover map) can help improve the spatial resolution of the estimated PM. In this study, several high-resolution geographical factors (i.e., land cover map, NDVI, road density, and population density) were exploited as the ancillary variates to maintain the spatial information. The space-based CV results show that the proposed framework performs well in the study area (e.g., $R^2$: 0.88 for $PM_{2.5}$ and 0.83 for $PM_{10}$), indicating that GEOS-FP data did not introduce large uncertainties. Meanwhile, the comparison for the spatial distribution (see an example in Figure 1) also signifies that the spatial resolution of the estimated PM is much higher than that of GEOS-FP data. Moreover, some relevant works that estimated ambient concentrations of air pollutants have employed coarse-resolution datasets as major predictors, such as [3-5]. In conclusion, the adoption of GEOS-FP datasets is justified in our study.

[Figure]

Figure 1: Daily (20190101) spatial distribution of the estimated $PM_{2.5}$ and GEOS-FP data (SO4CMASS, BCCMASS, and NICMASS)

**References:**

[1] Yang, Q., Yuan, Q., Li, T., & Yue, L. (2020). Mapping PM2. 5 concentration at high resolution using a cascade random forest based downscaling model: Evaluation and application. Journal of Cleaner Production, 277, 123887.

[2] Yang, Q., Yuan, Q., Yue, L., Li, T., Shen, H., & Zhang, L. (2020). Mapping PM2. 5 concentration at a sub-km level resolution: A dual-scale retrieval approach. ISPRS Journal of Photogrammetry and Remote Sensing, 165, 140-151.

[3] Zhan, Y., Luo, Y., Deng, X., Zhang, K., Zhang, M., Grieneisen, M. L., & Di, B. (2018). Satellite-based estimates of daily NO2 exposure in China using hybrid random forest and spatiotemporal kriging model. Environmental science & technology, 52(7), 4180-4189.

[4] Liu, D., Di, B., Luo, Y., Deng, X., Zhang, H., Yang, F., ... & Yu, Z. (2019). Estimating ground-level CO concentrations across China based on the national monitoring network and MOPITT: potentially overlooked CO hotspots in the Tibetan Plateau. Atmospheric Chemistry and Physics, 19(19), 12413-12430.

[5] Chen, Z. Y., Zhang, R., Zhang, T. H., Ou, C. Q., & Guo, Y. (2019). A kriging-calibrated machine learning method for estimating daily ground-level NO2 in mainland China. Science of The Total Environment, 690, 556-564.

**Minor comments:**

**Q2: The GEOS-FP provides more than described variables for use. Why do you choose the column mass density variable rather than others, say surface concentration variables?**

**Response:** Thank you for your comment. The estimation accuracy for the proposed framework using column mass density variables is slightly better than that using surface concentration variables. Hence, column mass density variables are adopted in our study.

**Q3: Line 279-280, because you are modeling on the same dependent variable (i.e. PM concentrations), RMSE is comparable though you can choose not to describe for conciseness.**

**Response:** Thank you for your comment. In our study, the numbers of matched samples are provided in Figure S9. Since missing values exist in the VIIRS DB AOD product, the data distributions of the estimated results through the proposed framework and AOD-based are different. For instance, the estimated results through the AOD-based is probably available for tens of days in a year, which cannot represent the annual condition. As for a matched grid, if the estimated results of $PM_{2.5}$ through the AOD-based is mainly valid in DJF/JJA, their RMSE could be relatively large/small. As a result, the comparison for RMSE is likely inappropriate and was not discussed in the manuscript.

**Q4: Line 314, larger => smaller?**

**Response:** Thank you for your comment. This statement has been revised in the manuscript.

---

## Author Comment (AC2) · 30 Nov 2020

**Response to Comments on the Manuscript (acp-2020-1004):**

**"Estimating daily full-coverage and high-accuracy 5-km ambient particulate matters across China: considering their precursors and chemical compositions"**
* * *
**Response to Comments of Referee #1:**

**General comment:**

**Wang et al. present an analysis of particulate matter over China. They use a Light Gradient Boosting Machine regression technique to combine satellite observations and model simulations to estimate surface PM 2.5 and PM 10. While the broad topics addressed in this manuscript (i.e. pollutant estimation, data-model fusion, machine learning, etc.) are important areas of research, I cannot recommend this paper for publication due to serious methodological issues and the fact that very similar and more comprehensive work has already been published.**

**Response:** We would like to take this opportunity to gratefully thank the referee for his/her comments and recommendations for improving the paper. An item-by-item response to the interesting comments raised by the referee follows. Thanks for your time.

**Similarity to previous works:**

**Van Donkelaar et al. (2019) and Hammer et al. (2020) use a far simpler statistical method to achieve similar performance at across larger temporal, spatial, and chemical scales. As a specific example, Hammer et al. (2020) use a relatively simple linear adjustment of satellite observations to achieve similar performance at similar resolution globally for 20 years.**

**Response:** Thank the referee for his/her important comments. The mentioned works (Van Donkelaar et al., 2019; Hammer et al., 2020) provided valuable results and made a significant contribution to the

scientific community. However, the proposed study is not similar to these works and many distinctions/highlights can be found in our study compared to them, which are described as follows.

1) The purpose of our study is different from those of them. The proposed study aims at estimating the daily full-coverage PM using the datasets of their precursors & chemical compositions instead of AOD products. In our study, the validation results signify that the estimation model can perform well without the input of AOD. By contrast, the mentioned works adopted numerous AOD products (e.g., MAIAC, DB, and MISR) as major variates. To be specific, see "We use **AOD** retrieved from radiances measured by four satellite instruments..." in Hammer et al. (2020) and "We combined **AOD** from multiple satellite products…" in Van Donkelaar et al. (2019). As for chemical species, these works employed a CTM (GEOS-Chem), while our study utilizes the datasets from two sources: remote sensing (S5P-TROPOMI) and data assimilation (GEOS-FP). These indicate that the intention (or emphasis) of the proposed study entirely differs from these works.

2) The temporal resolution of the estimated PM in the proposed study (daily) is much higher than those in the mentioned works (annual). To be specific, see "The temporal resolution of these globally fused $PM_{2.5}$ estimates focused on **annual** mean values to inform global health assessments…" in Hammer et al. (2020) and " **Annual** $PM_{2.5}$ composition estimates resulting from this effort…" in Van Donkelaar et al. (2019).

3) Only the annual estimated results of the mentioned works were validated at a global scale (Hammer et al., 2020) or over North America (Van Donkelaar et al., 2019). Meanwhile, the ground-based sites over China were not selected for an individual validation in these works. By contrast, the daily estimated PM are validated across China with multiple methods in our study. As a consequence, it cannot be concluded that the mentioned works achieved similar performance.

4) These works exploited an empirical method and the widely used GWR. The specific steps of these works (see their Supplement Information) are complicated and include plenty of data preprocessing procedures. By contrast, the proposed study adopts a convenient end-to-end approach based on an advanced gradient boosting algorithm (i.e., Light-GBM).

5) The motioned works only considered the estimation of ambient $PM_{2.5}$. In our study, ambient $PM_{2.5}$ and $PM_{10}$ are both estimated with high accuracy through the proposed approach.

**References:**

Hammer, M. S., van Donkelaar, A., Li, C., Lyapustin, A., Sayer, A. M., Hsu, N. C., ... & Brauer, M. (2020). Global Estimates and Long-Term Trends of Fine Particulate Matter Concentrations (1998-2018). Environmental Science & Technology.

Van Donkelaar, A., Martin, R. V., Li, C., & Burnett, R. T. (2019). Regional estimates of chemical composition of fine particulate matter using a combined geoscience-statistical method with information from satellites, models, and monitors. Environmental science & technology, 53(5), 2595-2611.

**Methodological Issues:**

**Section 3.1:** **This data processing is inappropriate for the input data as described in the supplement. In particular, downscaling wind fields through bicubic interpolation does not preserve mass or energy, nor does it accurately reproduce any higher order variability in the wind fields. More advanced statistical approaches (e.g. Kirchmeier et al., 2014) are necessary to produce physically consistent informations.**

**Response:** Thank the referee for his/her valuable comments. The resampling method is an important part of the data preprocessing steps. The recommended resampling method (Kirchmeier et al., 2014) is advanced and can preserve mass or energy. However, it requires ground-based stations of wind fields to acquire the Probability Density Function (PDF). Since the historical ground measurements of meteorological factors are undisclosed in China, this method is difficult to be employed. For previous related works about the estimation of PM, they usually adopted some simple methods to resample meteorological factors, such as the nearest neighbor method (Hu et al., 2014, 2017; Yao et al., 2019) and bilinear/bicubic/inverse distance weighted interpolation (Chen et al., 2019; Li et al., 2017; Wei et al., 2019; Yang et al., 2020; Shen et al., 2018; Ma et al., 2019). Therefore, the bicubic interpolation is selected as the resampling method in our study and the validation results do not suggest that the adoption of it will introduce large errors. At present, the researches about the influence of the errors from resampling methods on the estimated PM are scarce. This is a topic that is worth exploring and we will study it in our future works.

**References:**

Chen, Z. Y., Zhang, T. H., Zhang, R., Zhu, Z. M., Yang, J., Chen, P. Y., ... & Guo, Y. (2019). Extreme gradient boosting model to estimate PM2.5 concentrations with missing-filled satellite data in China. Atmospheric environment, 202, 180-189.

Hu, X., Belle, J. H., Meng, X., Wildani, A., Waller, L. A., Strickland, M. J., & Liu, Y. (2017). Estimating PM2. 5

concentrations in the conterminous United States using the random forest approach. Environmental science & technology, 51(12), 6936-6944.

Hu, X., Waller, L. A., Lyapustin, A., Wang, Y., Al-Hamdan, M. Z., Crosson, W. L., ... & Liu, Y. (2014). Estimating ground-level PM2. 5 concentrations in the Southeastern United States using MAIAC AOD retrievals and a two-stage model. Remote Sensing of Environment, 140, 220-232.

Li, T., Shen, H., Yuan, Q., Zhang, X., & Zhang, L. (2017). Estimating ground‐level PM2.5 by fusing satellite and station observations: a geo‐intelligent deep learning approach. Geophysical Research Letters, 44(23), 11-985.

Ma, Z., Liu, R., Liu, Y., & Bi, J. (2019). Effects of air pollution control policies on PM 2.5 pollution improvement in China from 2005 to 2017: a satellite-based perspective. Atmospheric Chemistry and Physics, 19(10), 6861-6877.

Shen, H., Li, T., Yuan, Q., & Zhang, L. (2018). Estimating regional ground‐level PM2. 5 directly from satellite top‐of‐atmosphere reflectance using deep belief networks. Journal of Geophysical Research: Atmospheres, 123(24), 13-875.

Wei, J., Huang, W., Li, Z., Xue, W., Peng, Y., Sun, L., & Cribb, M. (2019). Estimating 1-km-resolution PM2. 5 concentrations across China using the space-time random forest approach. Remote Sensing of Environment, 231, 111221.

Yang, Q., Yuan, Q., Yue, L., Li, T., Shen, H., & Zhang, L. (2020). Mapping PM2. 5 concentration at a sub-km level resolution: A dual-scale retrieval approach. ISPRS Journal of Photogrammetry and Remote Sensing, 165, 140-151.

Yao, F., Wu, J., Li, W., & Peng, J. (2019). A spatially structured adaptive two-stage model for retrieving ground-level PM2. 5 concentrations from VIIRS AOD in China. ISPRS Journal of Photogrammetry and Remote Sensing, 151, 263-276.

**Section 3.3:** **The 10x cross validation scheme described in this section does not account for the very large amount of spatial and temporal correlation present in environmental data. This will substantially and inappropriately bias the estimated skill of the machine learning model (e.g. Hastie et al. 2001, Roberts et al., 2017). The authors should perform a more rigorous evaluation of the model skill consistent with the standard in the field (e.g. Barnes et al., 2020). For example, the authors could cross validate through block methods by removing longer time periods of data or removing entire spatial regions of data beyond the autocorrelation length scale.**

**Response:** Thank the referee for his/her significant comments. The 10x cross validation scheme was widely used in previous related works about the estimation of PM over China (Chen et al., 2018a, 2018b; Wei et al., 2019; He et al., 2018; Zhang et al., 2019; Fang et al., 2016; Ma et al., 2014, 2016, 2019). The mentioned works (Van Donkelaar et al., 2019; Hammer et al., 2020) also utilized the 10x cross validation scheme. For instance, see "Performance was evaluated using a **10-fold** cross-validation…" in Van Donkelaar et al. (2019) and "The scatterplot shows **10-fold** out-of-sample 10% cross validation at sites..." in Hammer et al. (2020). In our study, a total of three 10x cross validation

schemes (i.e., sample-based, space-based, and time-based) are adopted. Among them, the space-based cross validation scheme is the most commonly used to evaluate the spatial accuracy of the estimated results (Li et al., 2020). Since we need to compare the validation results with the related works, these cross validation schemes are necessary in our study. The recommended methods (region-based or historical validation) are occasionally employed to verify the spatial or temporal predictive ability of the model, such as in Li et al. (2017) and Wei et al. (2019). At present, our study does not focus on the improvement to the predictive ability of the model. However, the historical validation results (removing longer time periods of data) are listed in Table r1 to present the temporal predictive ability of the proposed approach. For comparison, the historical validation results of some related works (Wei et al., 2019; Ma et al., 2016; He et al., 2018) are also provided in Table r2. As can be seen, the temporal predictive ability of the proposed approach is desired compared to these works.

Table r1: Metrics of the historical validation results in our study.

| Type | Training period | Validation period | Approach | $R^2$ | RMSE ($\mu g/m^3$) |
|------|-----------------|-------------------|----------|-------|--------------------|
| PM$_{2.5}$ | 2018.06.01-2019.05.31 (365) | 2019.06.01-2020.03.31 (305) | Proposed | 0.59 | 21.28 |
| | | | AOD-based | 0.54 | 22.37 |
| PM$_{10}$ | | | Proposed | 0.5 | 36.82 |
| | | | AOD-based | 0.42 | 48.12 |

Table r2: Metrics of the historical validation results in previous related works over China.

| Type | Reference | $R^2$ | RMSE ($\mu g/m^3$) | Validation period | Full-coverage |
|------|-----------|-------|--------------------|-------------------|---------------|
| PM$_{2.5}$ | Wei et al., 2019 | 0.55 | 27.38 | 2016 (366) | |
| | He et al., 2018 | 0.47 | 37.57 | 2014 (365) | False |
| | Ma et al., 2016 | 0.41 | - | 2014.01.01-2014.06.30 (181) | |

**References:**

Chen, G., Li, S., Knibbs, L. D., Hamm, N. A., Cao, W., Li, T., ... & Guo, Y. (2018b). A machine learning method to estimate PM2. 5 concentrations across China with remote sensing, meteorological and land use information. Science of the Total Environment, 636, 52-60.

Chen, G., Wang, Y., Li, S., Cao, W., Ren, H., Knibbs, L. D., ... & Guo, Y. (2018a). Spatiotemporal patterns of PM10 concentrations over China during 2005–2016: A satellite-based estimation using the random forests approach. Environmental Pollution, 242, 605-613.

Fang, X., Zou, B., Liu, X., Sternberg, T., & Zhai, L. (2016). Satellite-based ground PM2.5 estimation using timely structure adaptive modeling. Remote Sensing of Environment, 186, 152-163.

Hammer, M. S., van Donkelaar, A., Li, C., Lyapustin, A., Sayer, A. M., Hsu, N. C., ... & Brauer, M. (2020). Global Estimates and Long-Term Trends of Fine Particulate Matter Concentrations (1998-2018). Environmental Science & Technology.

He, Q., & Huang, B. (2018). Satellite-based mapping of daily high-resolution ground PM2. 5 in China via space-time regression modeling. Remote Sensing of Environment, 206, 72-83.

Li, T., Shen, H., Yuan, Q., Zhang, X., & Zhang, L. (2017). Estimating ground‐level PM2. 5 by fusing satellite and station observations: a geo‐intelligent deep learning approach. Geophysical Research Letters, 44(23), 11-985.

Li, T., Shen, H., Zeng, C., & Yuan, Q. (2020). A Validation Approach Considering the Uneven Distribution of Ground Stations for Satellite-Based PM 2.5 Estimation. IEEE Journal of Selected Topics in Applied Earth Observations and Remote Sensing, 13, 1312-1321.

Ma, Z., Hu, X., Huang, L., Bi, J., & Liu, Y. (2014). Estimating ground-level PM2. 5 in China using satellite remote sensing. Environmental science & technology, 48(13), 7436-7444.

Ma, Z., Hu, X., Sayer, A. M., Levy, R., Zhang, Q., Xue, Y., ... & Liu, Y. (2016). Satellite-based spatiotemporal trends in PM2. 5 concentrations: China, 2004–2013. Environmental health perspectives, 124(2), 184-192.

Ma, Z., Liu, R., Liu, Y., & Bi, J. (2019). Effects of air pollution control policies on PM 2.5 pollution improvement in China from 2005 to 2017: a satellite-based perspective. Atmospheric Chemistry and Physics, 19(10), 6861-6877.

Van Donkelaar, A., Martin, R. V., Li, C., & Burnett, R. T. (2019). Regional estimates of chemical composition of fine particulate matter using a combined geoscience-statistical method with information from satellites, models, and monitors. Environmental science & technology, 53(5), 2595-2611.

Wei, J., Huang, W., Li, Z., Xue, W., Peng, Y., Sun, L., & Cribb, M. (2019). Estimating 1-km-resolution PM2.5 concentrations across China using the space-time random forest approach. Remote Sensing of Environment, 231, 111221.

Zhang, T., Zang, L., Wan, Y., Wang, W., & Zhang, Y. (2019). Ground-level PM2. 5 estimation over urban agglomerations in China with high spatiotemporal resolution based on Himawari-8. Science of the total environment, 676, 535-544.

**Section 4.4:** **The feature importance results in this section call in question the validity of the modelling approach here. The top ranked variables include those that were unphysically downscaled: wind speeds (U, V, and Ustar) and turbulent evaporation, as well as total column ozone. As the vast majority of the ozone within a total column is present in the stratosphere, any surface predictive capacity associated with that variable is dubious. Additionally, the high importance ranking of $NO_2$ potentially indicates that the machine learning model is predicting based simply off the proximity to combustion sources. This explains why the model performance is so poor over more remote regions in Western China and limits the applicability of the method developed.**

**Response:** Thank the referee for his/her meaningful comments. The total comments can be divided into three parts, which are replied as follows.

1) "unphysically downscaled meteorological factors (e.g., wind speed)": The reasons for the adoption of bicubic interpolation in our study have been explained above. Previous related works about the estimation of PM usually applied some simple methods to resample meteorological factors. There is no indication that bicubic interpolation will lead to large errors in our study.

2) "total column ozone": For the stratospheric $O_3$, a latest study (Chen et al., 2020) has shown that the downward transport of $O_3$ stemming from the stratosphere-to-troposphere exchange can be a significant contributor to background $O_3$. Such enhancement of background $O_3$ will affect ambient PM. In addition, ambient $O_3$ pollution is rapidly increasing over China in recent years (Liu et al., 2020; Wang et al., 2020) and the proportion of it may also rise in the total $O_3$ column. It is inferred that more surface information can be extracted from the total $O_3$ column in China compared to other regions. At present, the total $O_3$ column has been used to estimate ambient $O_3$ over China (Liu et al., 2020) and Tibetan Plateau (Li et al., 2020a), suggesting its surface predictive capacity. In China, ambient PM is associated with ambient $O_3$ (Chen et al., 2019). Therefore, the total $O_3$ column is introduced as an auxiliary variate in our study and the feature importance of it is ranked 9[th] and 7[th] for $PM_{2.5}$ and $PM_{10}$, respectively.

3) "poor model performance in Western China": The poor performance of the proposed approach in Western China primarily results from the imbalanced matched samples. Since most of the ground-based sites distribute in the populous regions, the matched samples are small in Western China. This makes it difficult to extract useful information over these regions. In our study, the high importance ranking of $NO_2$ potentially signifies that nitrate is generally the major contribution to ambient PM in China according to the current distribution of ground-based sites. Ambient PM presents strong heterogeneous spatial patterns over China (Li et al., 2017, 2020b); consequently, the model performance is spatially various due to the imbalanced matched samples. In previous related works about the estimation of PM over China, this phenomenon is common. Some examples (Chen et al., 2019; Wei et al., 2019; Li et al., 2020b) are provided in Figure r1-r3. At present, the purpose of our study does not aim at addressing the issue caused by the imbalanced (or small) matched samples. We will study it in our future works.

[Figure]

Figure r1: Distribution of the (a) sample-based CV $R^2$ and (b) site-based CV $R^2$ of the GTW-GRNN model (Li et al., 2020b).

[Figure]

Figure r2: Spatial distribution of the spatial cross-validation result (blue color indicates a better fit) with each grid with $PM_{2.5}$ monitors (Chen et al., 2019).

[Figure]

Figure r3: Spatial distributions of $R^2$ between $PM_{2.5}$ estimations and measurements from 2016 in China. Results are from the sample-based 10-cross-validation (Wei et al., 2019).

**References:**

Chen, L., Xing, J., Mathur, R., Liu, S., Wang, S., & Hao, J. (2020). Quantification of the enhancement of PM2.5 concentration by the downward transport of ozone from the stratosphere. Chemosphere, 126907.

Chen, J., Shen, H., Li, T., Peng, X., Cheng, H., & Ma, C. (2019). Temporal and Spatial Features of the Correlation between PM2.5 and O3 Concentrations in China. International Journal of Environmental Research and Public Health, 16(23), 4824.

Chen, Z. Y., Zhang, T. H., Zhang, R., Zhu, Z. M., Yang, J., Chen, P. Y., ... & Guo, Y. (2019). Extreme gradient boosting model to estimate PM2. 5 concentrations with missing-filled satellite data in China. Atmospheric environment, 202, 180-189.

Li, T., Shen, H., Yuan, Q., Zhang, X., & Zhang, L. (2017). Estimating ground‑level PM2. 5 by fusing satellite and station observations: a geo‑intelligent deep learning approach. Geophysical Research Letters, 44(23), 11-985.

Li, R., Zhao, Y., Zhou, W., Meng, Y., Zhang, Z., & Fu, H. (2020a). Developing a novel hybrid model for the estimation of surface 8 h ozone (O3) across the remote Tibetan Plateau during 2005–2018. Atmospheric Chemistry and Physics, 20(10), 6159-6175.

Li, T., Shen, H., Yuan, Q., & Zhang, L. (2020b). Geographically and temporally weighted neural networks for satellite-based mapping of ground-level PM2. 5. ISPRS Journal of Photogrammetry and Remote Sensing, 167, 178-188.

Liu, R., Ma, Z., Liu, Y., Shao, Y., Zhao, W., & Bi, J. (2020). Spatiotemporal distributions of surface ozone levels in China from 2005 to 2017: A machine learning approach. Environment International, 142, 105823.

Shao, Y., Ma, Z., Wang, J., & Bi, J. (2020). Estimating daily ground-level PM2. 5 in China with random-forest-based spatiotemporal kriging. Science of The Total Environment, 740, 139761.

Wang, Y., Wild, O., Chen, X., Wu, Q., Gao, M., Chen, H., ... & Wang, Z. (2020). Health impacts of long-term ozone exposure in China over 2013–2017. Environment International, 144, 106030.

Wei, J., Huang, W., Li, Z., Xue, W., Peng, Y., Sun, L., & Cribb, M. (2019). Estimating 1-km-resolution PM2.5 concentrations across China using the space-time random forest approach. Remote Sensing of Environment, 231, 111221.

**Figures: All figures with maps in this paper violate the ACP maps policy: https://www.atmospheric-chemistry-and-physics.net/submission.html#mapsaerials. "In order to depoliticize scientific articles, authors should avoid the drawing of borders or use of contested topographical names." The inset of the South China Sea does not aid in the scientific interpretation of the results presented in this manuscript in any way, and only confuses the figures.**

**Response:** Thank the referee for his/her comment. These figures have been redrawn in the manuscript.

**An example of revision is as follows:**

[Figure]

Figure 1. The spatial distribution of the ground-based stations over China. The base-map is the true color image of MODIS.

**Specific Comments:**

**Q1.1: L14 "Most of the existing approaches for the estimation of PM 2.5 and PM 10 employed the remote sensing Aerosol Optical Depth (AOD) products as the main variate." I don't believe this is the case. Most approaches to estimate PM 2.5 and PM 10 come from in situ observations of aerosol mass and size distributions.**

**Response:** Thank the referee for his/her comment. This statement is inaccurate and has been reworded in the manuscript.

**The main revision is as follows:**

At present, most of remote sensing based approaches for the estimation of $PM_{2.5}$ and $PM_{10}$ employed Aerosol Optical Depth (AOD) products as the main variate.

**Q1.2: L29 "conducive to the researches". I'm not sure what this sentence means.**

**Response:** Thank the referee for his/her comment. This statement has been reworded in the manuscript.

**The main revision is as follows:**

It is concluded that the full-coverage estimated results from our study will be helpful in the field of large-scale PM$_{2.5}$ and PM$_{10}$ monitoring over the regions where the AOD values are missing.

**Q1.3: L38 Satellite observations are much more expensive than ground-based monitoring.**

**Response:** Thank the referee for his/her comment. This statement is intended to express that the observations from existing atmospheric satellites (e.g., Terra, Himawari-8, and Suomi-NPP) can be adopted. We have deleted it in the manuscript.

**Q1.4: L45 This statement needs appropriate citation.**

**Response:** Thank the referee for his/her comment. This statement is inaccurate and has been reworded in the manuscript.

**The main revision is as follows:**

With regard to CTMs, the uncertainties of the emission inventories could be large in some areas (Li et al., 2017b) and it will consume time and energy to collect the necessary information for simulation (Chu et al., 2016). The approaches based on remote sensing satellites have been greatly developed in recent years (Sorek-Hamer et al., 2020).

**References:**

Sorek-Hamer, M., Chatfield, R., & Liu, Y. (2020). Strategies for using satellite-based products in modeling PM2.5 and short-term pollution episodes. Environment International, 144, 106057.

Chu, Y., Liu, Y., Li, X., Liu, Z., Lu, H., Lu, Y., ... & Liu, F. (2016). A review on predicting ground PM2.5 concentration using satellite aerosol optical depth. Atmosphere, 7(10), 129.

**Q1.5: Section 2.2 This section is far too lacking in details to interpret or reproduce the work presented in this paper. The authors should explicitly state the variables used in the main body of the manuscript.**

**Response:** Thank the referee for his/her comment. Since numerous variates (30) are introduced in the proposed approach, the specific description of them was considered tedious. To be more clear, the explicit statement about the variates used in our study has been appended in Section 3.2, following the

estimation function.

The general scheme of the model for estimating the ambient concentrations of PM$_{2.5}$ and PM$_{10}$ can be expressed as Eq. (1).

$$C_{PM} = f(VM_P, VM_{CC}, VA_{O3}, VA_{MF}, VA_{GF}, DOY) \tag{1}$$

where $C_{PM}$ signifies the estimated ambient concentrations of PM$_{2.5}$ and PM$_{10}$. $f$ denotes the estimation function for the ambient concentrations of PM$_{2.5}$ and PM$_{10}$ based on LGBM. $VM_P$ and $VM_{CC}$ indicate the main variates of the precursors and chemical compositions, respectively, for PM$_{2.5}$ and PM$_{10}$. $VA_{O3}$, $VA_{MF}$, and $VA_{GF}$ represent the auxiliary variates of the O$_3$ from TROPOMI, meteorological factors, and geographical factors, respectively; $DOY$ is the day of year. To be specific, $VM_P$ consists of nitrogendioxide_tropospheric_column and sulfurdioxide_total_vertical_column_1km; $VM_{CC}$ includes Black Carbon Column Mass Density, Organic Carbon Column Mass Density, Nitrate Column Mass Density, SO4 Column Mass Density, Dust Column Mass Density, Ammonium Column Mass Density, and Sea Salt Column Mass Density; $VM_{MF}$ covers 10-meter Specific Humidity, 10-meter Air Temperature, 10-meter Eastward Wind, 10-meter Northward Wind, Total Precipitable Water Vapor, Pbltop Pressure, Surface Pressure, Planetary Boundary Layer Height, Air Density at Surface, Surface Velocity Scale, and Evaporation from Turbulence; and $VA_{GF}$ contains 1_km_16_days_NDVI, the fractions of forest, savanna, grassland, cropland, urban, and aridland, road density, and population density.

**Q1.6: L120 This link does not work for me.**

**Response:** Thank the referee for his/her comment. This website requires the Microsoft Silverlight (> 4.0) (https://www.microsoft.com/getsilverlight/get-started/install/default) and the screenshot of it has been presented in Figure r4. In addition, the air quality data is also available at http://www.cnemc.cn/.

[Figure]

Figure r4: Screenshot of http://106.37.208.233:20035.

**Q1.7: Section 3.2 How were the hyperparameters selected? Was there any optimization or search algorithm applied here?**

**Response:** Thank the referee for his/her comment. Since the matched samples (e.g., 31*742932 for $PM_{2.5}$) are large and the training procedure requires plenty of time, the random search based on cross validation is adopted in our study for some key hyperparameters (e.g., num_leaves and learning_rate).

**Q1.8: Figure 5 The colors being split into only 5 bins makes assessing performance difficult. Consider using a continuous colorbar.**

**Response:** Thank the referee for his/her comment. These figures have been redrawn in the manuscript.

**An example of revision is as follows:**

[Figure]

Figure 5. The spatial distribution of $R^2$s for the space-based CV at each matched grid over China. The black crosses denote that the significance levels (p) of the metrics are not less than 0.01 at these matched grids.

**Q1.9: L329 The authors should explicitly show evidence for this incorrect estimation through sampling.**

**Response:** Thank the referee for his/her comment. The annual validation results (space-based cross validation) of the estimated results (proposed and AOD-based) in 2019 over China are depicted in Figure r5. As can be seen, the annual performance of AOD-based is poor (large bias) due to the sampling that discards the missing values in the AOD product, with RPEs of 28.07% and 33.62% for $PM_{2.5}$ and $PM_{10}$, respectively. By contrast, the proposed approach performs well for the annual estimation.

[Figure]

Figure r5: The density scatter plots of the annual validation results for $PM_{2.5}$ and $PM_{10}$ in 2019 over China. The black solid line signifies the fitted line and the color bar denotes the density of samples. Y: annual estimated ambient concentrations of $PM_{2.5}$ and $PM_{10}$; X: annual ground-based ambient concentrations of $PM_{2.5}$ and $PM_{10}$.

---

## Referee Comment (RC2) · Anonymous Referee #2 · 1 Feb 2021

**General comments**

In this paper, Wang et al. proposed a framework to estimate the daily PM2.5 and PM10 concentration over China by combining multiple data sources with a Light Gradient Boosting Machine learning method. They included satellite product (TROPOMI) and modeled data assimilation dataset (GEOS-FP) as main predictors. Even though they showed some reasonable statistics from the validation process, the advantages of this method are not well justified. Also, some flaws are found in their validation process. As a result, I do not recommend this paper for publication in ACP.

[Figure]

Major comments:

1. The selection of predicting variables from TROPOMI is too arbitrary and lacks justification. In this paper, the authors only considered column N0x and SO2 observations, which are the precursors of sulfate and nitrate. Both of them are large components of PM2.5. However, other components are also important. For example organic aerosols. Why the precursors of organic aerosols were not chosen as input predictors? Also, this idea of "PM is associated with ozone, so choose column ozone as one of the predictors" needs more justifications.

2. By using GEOS-FP, this method loses its flexibility to adjust its input conditions. GEOS-FP is one of the data assimilation products from GMAO. It is based largely on the model output of GEOS. The method in this paper is largely based on this dataset. According to their fig 6, 4 out of 6 top feature importance for their PM2.5 prediction are from GEOS-FP. What this means is that their prediction is mostly controlled by a dataset that they could not make any modifications to. For example, the GEOS-FP system has been updated to version 5.25 in January 2020 (see this link https://gmao.gsfc.nasa.gov/news/geos_system_news/2020/GEOS_FP_upgrade_5_25_1.php ). Let alone some discontinuous issues posed by all the updates along with the release of each GEOS-FP products. On the other hand, the authors argue that previous studies using CTMs have the limitation of large uncertainty in the emission inventories of the CTMs. By saying that, GEOS-FP also has the same problem of large uncertainty in their emission inventories. By using CTMs instead of data assimilation datasets as prediction inputs, researchers can do their best to narrow the uncertainty in their model simulations and could also conduct sensitivity experiments. From this point of view, methods using CTMs or earth models (e.g. GEOS) would be better for this kind of prediction.

3. The validation process exists flows. First, the authors defined their "AOD-based" control estimate, which is using VIIRS AOD to replace the TROPOMI and GEOS-FP in their estimating framework. However, this is not the case of previous AOD based

studies, especially those most influential ones, for example, van Donkelaar et al. 2019. Previous AOD based studies usually combined with model simulations and ground-based measurements to best use the information of satellite AOD products. What has been done in the paper was that comparing an estimate from a missing data AOD product with an estimate from the combination of satellite observations and model output. So that it caused the issue of comparing apple with orange; moreover, the selling point of this paper is the daily PM2.5 estimate. So validation regarded to a daily resolution would be the most convincing. Otherwise, why not just estimate the seasonal or annual PM concentrations, which are more useful in current epidemiological studies. The validation results from the time-based cross-validation method were the worst according to their table1.

Specific comments

1. Line 99: is it necessary to mention that the area of this study was chosen because that China has the largest population in the world? Maybe rephrase this sentence or delete it.

2. Line 106: "monitor"? maybe use "estimate"

3. Line 120: any publications about the data validation, calibration, and uncertainty analysis of the Chinese PM2.5/PM10 measurements from CNEMC?

4. Follow point 3, the authors could try to use this method on other regions where have an extensive ground-based measurements network, for example, North America, to test their validity.

Technical corrections

1. Section index is wrong. For example, two "3.2" sections exist in the main text.

2. Line40: should be "van Donkelaar" instead of "Van Donkelaar"

3. Define "DUCMASS" in the main text.

[Figure]

References

Van Donkelaar, A, Randall V. Martin, Chi Li, and Richard T. Burnett. "Regional Estimates of Chemical Composition of Fine Particulate Matter Using a Combined Geoscience-Statistical Method with Information from Satellites, Models, and Monitors." Environmental Science & Technology 2019. https://doi.org/10.1021/acs.est.8b06392.
* * *

---

## Author Comment (AC3) · 26 Feb 2021

**Response to Comments on the Manuscript (acp-2020-1004):**

**"Estimating daily full-coverage and high-accuracy 5-km ambient particulate matters across China: considering their precursors and chemical compositions"**
* * *
**Response to Comments of Referee #2:**

**General comment:**

In this paper, Wang et al. proposed a framework to estimate the daily PM2.5 and PM10 concentration over China by combining multiple data sources with a Light Gradient Boosting Machine learning method. They included satellite product (TROPOMI) and modeled data assimilation dataset (GEOS-FP) as main predictors. Even though they showed some reasonable statistics from the validation process, the advantages of this method are not well justified. Also, some flaws are found in their validation process. As a result, I do not recommend this paper for publication in ACP.

**Response:** We would like to express our sincere gratitude to the referee for his/her comments and recommendations for improving the paper. An item-by-item response to the comments raised by the referee follows. Thanks for your time.

**Major comments:**

**Q2.1:** The selection of predicting variables from TROPOMI is too arbitrary and lacks justification. In this paper, the authors only considered column NOx and SO2 observations, which are the precursors of sulfate and nitrate. Both of them are large components of PM2.5. However, other components are also important. For example organic aerosols. Why the precursors of organic aerosols were not chosen as input predictors? Also, this idea of "PM is

**associated with ozone, so choose column ozone as one of the predictors" needs more justifications.**

**Response:** Thank the referee for his/her significant comments. The purpose of our study is to estimate daily full-coverage PM at high spatial resolution using the datasets of their precursors & chemical compositions instead of AOD products. Therefore, the selection of predicting variables should be carefully considered. As for remote sensing sensors, only TROPOMI can generate daily high-spatial-resolution (e.g., 5-km) and high-coverage chemical species at present. In our study, the atmospheric products of $NO_2$ and $SO_2$ from TROPOMI were adopted, regarded as two precursors of PM. However, there is merely one precursor of organic aerosol, i.e., formaldehyde, belonging to the atmospheric products from TROPOMI (see https://earth.esa.int/web/guest/missions/esa-eo-missions/sentinel-5p). Since formaldehyde is generally not a major organic aerosol precursor (Hallquist et al., 2009; Volkamer et al., 2006), this product was not utilized. By contrast, the Organic Carbon Column Mass Density (carbon-related) from GEOS-FP was used in this paper. Furthermore, some other chemical compositions of PM were also acquired from GEOS-FP, including the Nitrate Column Mass Density (nitrate-related), SO4 Column Mass Density (sulfate-related), Black Carbon Column Mass Density (carbon-related), Dust Column Mass Density (dust-related), Ammonium Column Mass Density (ammonium-related), and Sea Salt Column Mass Density (sea salt-related). All of these variables are employed according to the major chemical compositions of PM (i.e., nitrate, sulfate, carbon, dust, ammonium, and sea salt) (Baker et al., 2007; Tucker et al., 2000; Zheng et al., 2005; Wang et al., 2019; Pui et al., 2014). We have appended this statement in the manuscript. It is concluded that the selected variables are sufficient to estimate PM over China. The validation results show that the estimation model achieves a satisfactory performance (e.g., space-based CV $R^2$: 0.88 for $PM_{2.5}$ and 0.83 for $PM_{10}$) in our study, which also signify this point.

The justification for the adoption of total $O_3$ column is presented as follows. With regard to stratospheric $O_3$, a latest study (Chen et al., 2020) has shown that the downward transport of $O_3$ stemming from the stratosphere-to-troposphere exchange can be a significant contributor to background $O_3$. Such enhancement of background $O_3$ will affect ambient PM. In addition, ambient $O_3$ pollution is rapidly increasing over China in recent years (Liu et al., 2020; Wang et al., 2020) and the proportion of it may also rise in the total $O_3$ column. At present, the total $O_3$ column has been used to estimate ambient $O_3$ over China (Liu et al., 2020) and Tibetan Plateau (Li et al., 2020), suggesting its

surface predictive capacity. In China, ambient PM is associated with ambient $O_3$ (Chen et al., 2019). Therefore, the total $O_3$ column is introduced as an auxiliary variable in our study.

**The main revision is as follows:**

In our study, all of the chemical species are selected in accordance with the major component of $PM_{2.5}$ and $PM_{10}$ (i.e., nitrate, sulfate, carbon, dust, ammonium, and sea salt) (Baker et al., 2007; Tucker et al., 2000; Zheng et al., 2005; Wang et al., 2019; Pui et al., 2014).

**References:**

Baker, K. and Scheff, P.: Photochemical model performance for PM2.5 sulfate, nitrate, ammonium, and precursor species SO2, HNO3, and NH3 at background monitor locations in the central and eastern United States, Atmos. Environ., 41(29), 6185-6195, 2007.

Chen, L., Xing, J., Mathur, R., Liu, S., Wang, S., & Hao, J. (2020). Quantification of the enhancement of PM2.5 concentration by the downward transport of ozone from the stratosphere. Chemosphere, 126907.

Chen, J., Shen, H., Li, T., Peng, X., Cheng, H., & Ma, C. (2019). Temporal and Spatial Features of the Correlation between PM2.5 and O3 Concentrations in China. International Journal of Environmental Research and Public Health, 16(23), 4824.

Hallquist, M., Wenger, J. C., Baltensperger, U., Rudich, Y., Simpson, D., Claeys, M., ... & Wildt, J. (2009). The formation, properties and impact of secondary organic aerosol: current and emerging issues. Atmospheric chemistry and physics, 9(14), 5155-5236.

Li, R., Zhao, Y., Zhou, W., Meng, Y., Zhang, Z., & Fu, H. (2020). Developing a novel hybrid model for the estimation of surface 8 h ozone (O3) across the remote Tibetan Plateau during 2005–2018. Atmospheric Chemistry and Physics, 20(10), 6159-6175.

Liu, R., Ma, Z., Liu, Y., Shao, Y., Zhao, W., & Bi, J. (2020). Spatiotemporal distributions of surface ozone levels in China from 2005 to 2017: A machine learning approach. Environment International, 142, 105823.

Pui, D. Y., Chen, S. C., & Zuo, Z. (2014). PM2.5 in China: Measurements, sources, visibility and health effects, and mitigation. Particuology, 13, 1-26.

Tucker, W. G.: An overview of PM2.5 sources and control strategies, Fuel Process. Technol., 65, 379-392, 2000.

Volkamer, R., Jimenez, J. L., San Martini, F., Dzepina, K., Zhang, Q., Salcedo, D., ... & Molina, M. J. (2006). Secondary organic aerosol formation from anthropogenic air pollution: Rapid and higher than expected. Geophysical Research Letters, 33(17).

Wang, Y., Wild, O., Chen, X., Wu, Q., Gao, M., Chen, H., ... & Wang, Z. (2020). Health impacts of long-term ozone exposure in China over 2013–2017. Environment International, 144, 106030.

Wang, Y., Li, W., Gao, W., Liu, Z., Tian, S., Shen, R., and Song, T.: Trends in particulate matter and its chemical compositions in China from 2013–2017, Science China Earth Sciences., 62(12), 1857-1871, 2019.

Zheng, M., Salmon, L. G., Schauer, J. J., Zeng, L., Kiang, C. S., Zhang, Y., & Cass, G. R. (2005). Seasonal trends in PM2. 5 source contributions in Beijing, China. Atmospheric Environment, 39(22), 3967-3976.

**Q2.2:** By using GEOS-FP, this method loses its flexibility to adjust its input conditions. GEOS-FP is one of the data assimilation products from GMAO. It is based largely on the model output of GEOS. The method in this paper is largely based on this dataset. According to their fig 6, 4 out of 6 top feature importance for their PM2.5 prediction are from GEOS-FP. What this means is that their prediction is mostly controlled by a dataset that they could not make any modifications to. For example, the GEOS-FP system has been updated to version 5.25 in January 2020. Let alone some discontinuous issues posed by all the updates along with the release of each GEOS-FP products. On the other hand, the authors argue that previous studies using CTMs have the limitation of large uncertainty in the emission inventories of the CTMs. By saying that, GEOS-FP also has the same problem of large uncertainty in their emission inventories. By using CTMs instead of data assimilation datasets as prediction inputs, researchers can do their best to narrow the uncertainty in their model simulations and could also conduct sensitivity experiments. From this point of view, methods using CTMs or earth models (e.g. GEOS) would be better for this kind of prediction.

**Response:** Thank the referee for his/her valuable comments. To be clearer, the total comments are divided into two parts, which are replied as follows.

"By using GEOS-FP, this method loses its flexibility to adjust its input conditions. GEOS-FP is one of the data assimilation products from GMAO. It is based largely on the model output of GEOS. The method in this paper is largely based on this dataset. According to their fig 6, 4 out of 6 top feature importance for their PM2.5 prediction are from GEOS-FP. What this means is that their prediction is mostly controlled by a dataset that they could not make any modifications to. For example, the GEOS-FP system has been updated to version 5.25 in January 2020. Let alone some discontinuous issues posed by all the updates along with the release of each GEOS-FP products."

[Figure]

Figure r1:Daily (20190101) spatial distribution of the GEOS-FP, estimated, and ground-based PM$_{2.5}$. The circles represent the ground-based sites.

Apart from GEOS-FP, multiple datasets from other sources were adopted in our study, including

TROPOMI, MODIS, GPW, and OpenStreetMap. Especially, high-resolution geographical factors, such as NDVI, road density, and population density, could maintain the spatial information. In addition, an advance ensemble learning method, i.e., light gradient boosting machine, was exploited to fuse the multisource data using ground-truth values. Space-based CV results show that the proposed framework performs well in the study area (e.g., $R^2$: 0.88 for $PM_{2.5}$ and 0.83 for $PM_{10}$), suggesting that GEOS-FP data did not introduce large uncertainties. An example to compare the spatial distribution between the estimated and GEOS-FP $PM_{2.5}$ is demonstrated in Figure r1. The GEOS-FP $PM_{2.5}$ is calculated via this formula: $PM_{2.5}=1.375*SO_4+2.1*OC+BC+DS_{2.5}+SS_{2.5}$ (Xiao et al., 2018). As can be seen, the spatial patterns of the estimated $PM_{2.5}$ are much closer to the actual measurements than GEOS-FP, with a higher spatial resolution. Therefore, the estimation accuracy is greatly improved compared to GEOS-FP based on posterior techniques.

**"On the other hand, the authors argue that previous studies using CTMs have the limitation of large uncertainty in the emission inventories of the CTMs. By saying that, GEOS-FP also has the same problem of large uncertainty in their emission inventories. By using CTMs instead of data assimilation datasets as prediction inputs, researchers can do their best to narrow the uncertainty in their model simulations and could also conduct sensitivity experiments. From this point of view, methods using CTMs or earth models (e.g. GEOS) would be better for this kind of prediction."**

In this paper, the previous studies using CTMs refer to the works merely considering CTMs without other techniques. Our study does not focus on comparing to or arguing about the methods using CTMs. The mention of CTMs in the introduction was only to elicit that the approaches based on remote sensing satellites have been greatly developed in recent years. The statements about CTMs is confusing and has been rephrased in the manuscript. CTMs and GEOS-FP both potential present large uncertainty in their emission inventories. At present, the outputs of CTMs have been combined with other datasets (e.g., remote sensing) to estimate $PM_{2.5}$, such as the mentioned work (van Donkelaar et al., 2019). In our study, we also fused multiple datasets from new sources, including new remote sensing sensor (TROPOMI) and new data assimilation (GEOS-FP). Moreover, the present study is a novel attempt to estimate daily full-coverage PM based on the datasets of their precursors & chemical compositions instead of AOD products. The validation results show that the proposed framework can perform well

without the input of AOD in the study area.

**The main revision is as follows:**

With regard to CTMs, the uncertainties of the emission inventories could be large in some areas (Li et al., 2017b) and it will consume time and energy to collect the necessary information for simulation (Chu et al., 2016). The approaches based on remote sensing satellites have been greatly developed in recent years (Sorek-Hamer et al., 2020).

**References:**

Xiao, Q., Chang, H. H., Geng, G., & Liu, Y. (2018). An ensemble machine-learning model to predict historical PM2. 5 concentrations in China from satellite data. Environmental science & technology, 52(22), 13260-13269.

van Donkelaar, A., Martin, R. V., Li, C., & Burnett, R. T. (2019). Regional estimates of chemical composition of fine particulate matter using a combined geoscience-statistical method with information from satellites, models, and monitors. Environmental science & technology, 53(5), 2595-2611.

**Q2.3:** **The validation process exists flows. First, the authors defined their "AOD-based" control estimate, which is using VIIRS AOD to replace the TROPOMI and GEOS-FP in their estimating framework. However, this is not the case of previous AOD based studies, especially those most influential ones, for example, van Donkelaar et al. 2019. Previous AOD based studies usually combined with model simulations and ground-based measurements to best use the information of satellite AOD products. What has been done in the paper was that comparing an estimate from a missing data AOD product with an estimate from the combination of satellite observations and model output. So that it caused the issue of comparing apple with orange; moreover, the selling point of this paper is the daily PM2.5 estimate. So validation regarded to a daily resolution would be the most convincing. Otherwise, why not just estimate the seasonal or annual PM concentrations, which are more useful in current epidemiological studies. The validation results from the time-based cross-validation method were the worst according to their table1.**

**Response:** Thank the referee for his/her important comments. To fully validate the estimated results, a total of three CV schemes (i.e., sample-based, space-based, and time-based) were considered in our study. Meanwhile, all of the validation results were performed at daily resolution, including the overall validation (Figure 4), regional validation (Table 1), seasonal validation (Fig. S3-S6), and grid-based validation (Figure 5, S7-S9). The examples of daily estimated results in different seasons (20190122,

20190501, 20190803, and 20191111) were also provided in Figure 7 to show the daily spatial distributions.

Moreover, a baseline was devised in our study. Since a large number of related studies over China based on machine learning methods only adopted remote sensing AOD products (no model simulations) to estimate PM (Li et al., 2020; Yang et al., 2019; Wei et al., 2019; He et al., 2018, 2020; Yao et al., 2019; Ma et al., 2016; Chen et al., 2018a, 2018b, 2019; Wang et al., 2019; Zhang et al., 2019; Xue et al., 2020), the VIIRS DB AOD product was selected as the AOD-based. However, related studies could utilize various techniques to improve their estimation performance. For instance, Li et al. (2020) proposed GTWNN; Wei et al. (2019) developed STRF; Kong et al. (2020) combined model simulations and ground-based measurements. It is very difficult to duplicate their experiments as baselines due to different hardware facilities, large time consumptions, closed data sources, unspecified model parameters, etc. By contrast, we compared our metrics to those of related studies over China in recent years, which are listed in Table r1 (or see Table S5 in the supplementary materials). This is a common strategy used in previous studies for comparison with other works (Wei et al., 2019; Jiang et al., 2020; Kong et al., 2020). Apart from Kong et al. (2020) (model simulations), another two related studies over China using model simulations or reanalysis datasets have been appended in the table (Xue et al., 2017; Xiao et al., 2018), as the referee pointed out. The study area of the mentioned work (van Donkelaar et al., 2019) is North America and its temporal resolution is annual. Therefore, this work cannot be compared in our study.

Table r1: Detailed information about the previous related works over China. SACV: sample-based CV; SPCV: space-based CV; TICV: time-based CV; SR: spatial resolution; TR: temporal resolution; FC: full-coverage; T: true; F: false; MF: the factors which lead to the missing values in the estimated results.

| Type | Reference | Metric | SACV | SPCV | TICV | SR | TR | Study period | FC | MF |
|------|-----------|--------|------|------|------|-----|-----|--------------|-----|-----|
| PM$_{2.5}$ | Proposed | $R^2$ | 0.93 | 0.88 | 0.73 | 5-km | Daily | 2019※ | T | None |
| | | RMSE | 8.87 μg/m$^3$ | 11.56 μg/m$^3$ | 17.3 μg/m$^3$ | | | | | |
| | | RPE | 22.8% | 29.8% | 44.5% | | | | | |
| | Wei et al., 2019 | $R^2$ | 0.85 | 0.83 | 0.63 | 1-km | Daily | 2016 | F | Cloud, snow/ice |
| | | RMSE | 15.57 μg/m$^3$ | 16.63 μg/m$^3$ | 24.83 μg/m$^3$ | | | | | |
| | | RPE | - | - | - | | | | | |
| | He et al., 2018 | $R^2$ | 0.8 | | | 3-km | Daily | 2015 | F | Cloud, snow/ice, bright surface |
| | | RMSE | 18 μg/m$^3$ | - | - | | | | | |

| | | | | | | | | | |
|---|---|---|---|---|---|---|---|---|---|
| | Yao et al., 2019 | RPE | - | | | | | | |
| | | R² | | 0.6 | | | | | |
| | | RMSE | - | 21.76 μg/m³ | - | 6-km | Daily | 2014 | F | Cloud, snow/ice, bright surface |
| | | RPE | | - | | | | | |
| | Li et al., 2020 | R² | 0.8 | 0.79 | | | | | |
| | | RMSE | 17.38 μg/m³ | 17.81 μg/m³ | - | 10-km | Daily | 2015 | F | Cloud, snow/ice |
| | | RPE | 31.5% | 32.29% | | | | | |
| | Jiang et al., 2020 | R² | 0.85 | 0.74 | | | | | |
| | | RMSE | 11.02 μg/m³ | 14.65 μg/m³ | - | 1-km | Daily* | 2018.03.01-2019.02.28 | T | None |
| | | RPE | - | - | | | | | |
| | Kong et al., 2020 | R² | | 0.86 | | | | | |
| | | RMSE | - | 15.1 μg/m³ | - | 15-km | Daily* | 2013–2018 | T | None |
| | | RPE | | - | | | | | |
| | Xue et al., 2017 | R² | | 0.72 | | | | | |
| | | RMSE | - | 23 μg/m³ | - | 10-km | Daily | 2014 | T | None |
| | | RPE | | 41% | | | | | |
| | Xiao et al., 2018 | R² | 0.79 | 0.76 | 0.73 | 10-km | Daily | 2013-2016 | T | None |
| | | RMSE | - | - | - | | | | | |
| | | RPE | | | | | | | | |
| PM₁₀ | Proposed | R² | 0.91 | 0.84 | 0.67 | 5-km | Daily | 2019※ | T | None |
| | | RMSE | 16.92 μg/m³ | 22.03 μg/m³ | 31.33 μg/m³ | | | | | |
| | | RPE | 24.5% | 31.9% | 45.4% | | | | | |
| | Chen et al., 2018b | R² | | 0.78 | | | | | |
| | | RMSE | - | 31.54 μg/m³ | - | 10-km | Daily | 2005–2016 | F | Cloud, snow/ice |
| | | RPE | | - | | | | | |
| | Kong et al., 2020 | R² | | 0.81 | | | | | |
| | | RMSE | - | 28.8 μg/m³ | - | 15-km | Daily* | 2013–2018 | T | None |
| | | RPE | | - | | | | | |

Note:
1. The symbols of * represent that the works could provide the estimated results at various temporal resolutions, while the metrics listed in the table are computed from the daily estimation.
2. ※: Only the metrics computed from the estimated results through the proposed approach for a whole year (2019) are listed in the table to be fairly compared to previous works. The study period of this paper is from June 1, 2018 to March 31, 2020.

The validation results from the time-based CV were the worst since the temporal heterogeneity of PM is usually strong (Li et al., 2017, 2020). In other word, the temporal variations of PM could not be fully captured. This phenomenon can be discovered in previous related studies over China, such as

Wei et al. (2019) (time-based CV $R^2$: 0.63) and Xiao et al. (2018) (time-based CV $R^2$: 0.73). Compared to them, the time-based CV results in our study are acceptable.

In conclusion, it is believed that the validation process was justified in this paper.

**References:**

Chen, G., Li, S., Knibbs, L. D., Hamm, N. A., Cao, W., Li, T., ... & Guo, Y. (2018a). A machine learning method to estimate PM2. 5 concentrations across China with remote sensing, meteorological and land use information. Science of the Total Environment, 636, 52-60.

Chen, J., Yin, J., Zang, L., Zhang, T., & Zhao, M. (2019). Stacking machine learning model for estimating hourly PM2. 5 in China based on Himawari 8 aerosol optical depth data. Science of The Total Environment, 697, 134021.

Chen, G., Wang, Y., Li, S., Cao, W., Ren, H., Knibbs, L. D., and Guo, Y. (2018b). Spatiotemporal patterns of PM10 concentrations over China during 2005–2016: A satellite-based estimation using the random forests approach, Environmental Pollution., 242, 605-613.

He, Q., Gu, Y., & Zhang, M. (2020). Spatiotemporal trends of PM2. 5 concentrations in central China from 2003 to 2018 based on MAIAC-derived high-resolution data. Environment international, 137, 105536.

He, Q. and Huang, B. (2018): Satellite-based mapping of daily high-resolution ground PM2.5 in China via space-time regression modeling, Remote Sensing of Environment., 206, 72-83.

Jiang, T., Chen, B., Nie, Z., Zhehao, R., Xu, B., and Tang, S. (2020). Estimation of hourly full-coverage PM2.5 concentrations at 1-km resolution in China using a two-stage random forest model, Atmospheric Research., 105146.

Kong, L., Tang, X., Zhu, J., Wang, Z., Li, J., Wu, H., and Liu, B.: A Six-year long (2013–2018) High-resolution Air Quality Reanalysis Dataset over China base on the assimilation of surface observations from CNEMC. (2020). Earth System Science Data Discussions., 1-44.

Li, T., Shen, H., Yuan, Q., Zhang, X., & Zhang, L. (2017). Estimating ground‐level PM2. 5 by fusing satellite and station observations: a geo‐intelligent deep learning approach. Geophysical Research Letters, 44(23), 11-985.

Li, T., Shen, H., Yuan, Q., and Zhang, L. (2020). Geographically and temporally weighted neural networks for satellite-based mapping of ground-level PM2.5, ISPRS Journal of Photogrammetry and Remote Sensing., 167, 178-188.

Ma, Z., Hu, X., Sayer, A. M., Levy, R., Zhang, Q., Xue, Y., ... & Liu, Y. (2016). Satellite-based spatiotemporal trends in PM2. 5 concentrations: China, 2004–2013. Environmental health perspectives, 124(2), 184-192.

Van Donkelaar, A., Martin, R. V., Li, C., & Burnett, R. T. (2019). Regional estimates of chemical composition of fine particulate matter using a combined geoscience-statistical method with information from satellites, models, and monitors. Environmental science & technology, 53(5), 2595-2611.

Wang, W., Zhao, S., Jiao, L., Taylor, M., Zhang, B., Xu, G., & Hou, H. (2019). Estimation of PM2. 5 concentrations in China using a spatial back propagation neural network. Scientific reports, 9(1), 1-10.

Wei, J., Huang, W., Li, Z., Xue, W., Peng, Y., Sun, L., & Cribb, M. (2019). Estimating 1-km-resolution PM2. 5 concentrations across China using the space-time random forest approach. Remote Sensing of Environment, 231, 111221.

Xue, Y., Li, Y., Guang, J., Tugui, A., She, L., Qin, K., ... & Wang, Z. (2020). Hourly PM2. 5 Estimation over Central and Eastern China Based on Himawari-8 Data. Remote Sensing, 12(5), 855.

Xue, T., Zheng, Y., Geng, G., Zheng, B., Jiang, X., Zhang, Q., & He, K. (2017). Fusing observational, satellite remote sensing and air quality model simulated data to estimate spatiotemporal variations of PM2. 5 exposure in China. Remote Sensing, 9(3), 221.

Xiao, Q., Chang, H. H., Geng, G., & Liu, Y. (2018). An ensemble machine-learning model to predict historical PM2.5 concentrations in China from satellite data. Environmental science & technology, 52(22), 13260-13269.

Yang, L., Xu, H., & Jin, Z. (2019). Estimating ground-level PM2. 5 over a coastal region of China using satellite AOD and a combined model. Journal of Cleaner Production, 227, 472-482.

Yao, F., Wu, J., Li, W., and Peng, J. (2019). A spatially structured adaptive two-stage model for retrieving ground-level PM2.5 concentrations from VIIRS AOD in China, ISPRS Journal of Photogrammetry and Remote Sensing., 151, 263-276.

Zhang, T., Zang, L., Wan, Y., Wang, W., & Zhang, Y. (2019). Ground-level PM2. 5 estimation over urban agglomerations in China with high spatiotemporal resolution based on Himawari-8. Science of the total environment, 676, 535-544.

**Specific comments:**

**Q2.4:** **Line 99: is it necessary to mention that the area of this study was chosen because that China has the largest population in the world? Maybe rephrase this sentence or delete it.**

**Response:** Thank the referee for his/her comment. This statement has been removed in the manuscript.

**Q2.5:** **Line 106: "monitor"? maybe use "estimate".**

**Response:** Thank the referee for his/her comment. The word "monitor" has been replaced with "estimate" in the manuscript.

**The main revision is as follows:**

It is necessary to develop an approach that can estimate $PM_{2.5}$ and $PM_{10}$ across China continuously and precisely.

**Q2.6:** **Line 120: any publications about the data validation, calibration, and uncertainty analysis of the Chinese PM2.5/PM10 measurements from CNEMC?**

**Response:** Thank the referee for his/her comment. The required contents have been appended in the manuscript. The ground-based measurements from CNEMC have been widely used to estimate PM across China (Li et al., 2020; Yang et al., 2019; Wei et al., 2019; He et al., 2018, 2020; Yao et al., 2019; Ma et al., 2016; Chen et al., 2018a, 2018b, 2019; Wang et al., 2019; Zhang et al., 2019; Xue et al.,

2017, 2019, 2020; Kong et al., 2020), suggestion their reliability and accuracy.

**The main revision is as follows:**

The CNEMC can provide hourly ambient concentrations of $PM_{2.5}$ and $PM_{10}$ over China, which are obtained according to the technical specification of HJ 817-2018 (i.e., tapered element oscillating microbalance method or beta-attenuation method).

**References:**

Chen, G., Li, S., Knibbs, L. D., Hamm, N. A., Cao, W., Li, T., ... & Guo, Y. (2018a). A machine learning method to estimate PM2. 5 concentrations across China with remote sensing, meteorological and land use information. Science of the Total Environment, 636, 52-60.

Chen, J., Yin, J., Zang, L., Zhang, T., & Zhao, M. (2019). Stacking machine learning model for estimating hourly PM2. 5 in China based on Himawari 8 aerosol optical depth data. Science of The Total Environment, 697, 134021.

Chen, G., Wang, Y., Li, S., Cao, W., Ren, H., Knibbs, L. D., and Guo, Y. (2018b). Spatiotemporal patterns of PM10 concentrations over China during 2005–2016: A satellite-based estimation using the random forests approach, Environmental Pollution., 242, 605-613.

He, Q., Gu, Y., & Zhang, M. (2020). Spatiotemporal trends of PM2. 5 concentrations in central China from 2003 to 2018 based on MAIAC-derived high-resolution data. Environment international, 137, 105536.

He, Q. and Huang, B. (2018): Satellite-based mapping of daily high-resolution ground PM2.5 in China via space-time regression modeling, Remote Sensing of Environment., 206, 72-83.

Kong, L., Tang, X., Zhu, J., Wang, Z., Li, J., Wu, H., and Liu, B.: A Six-year long (2013–2018) High-resolution Air Quality Reanalysis Dataset over China base on the assimilation of surface observations from CNEMC. (2020). Earth System Science Data Discussions., 1-44.

Li, T., Shen, H., Yuan, Q., and Zhang, L. (2020). Geographically and temporally weighted neural networks for satellite-based mapping of ground-level PM2.5, ISPRS Journal of Photogrammetry and Remote Sensing., 167, 178-188.

Ma, Z., Hu, X., Sayer, A. M., Levy, R., Zhang, Q., Xue, Y., ... & Liu, Y. (2016). Satellite-based spatiotemporal trends in PM2. 5 concentrations: China, 2004–2013. Environmental health perspectives, 124(2), 184-192.

Wang, W., Zhao, S., Jiao, L., Taylor, M., Zhang, B., Xu, G., & Hou, H. (2019). Estimation of PM2. 5 concentrations in China using a spatial back propagation neural network. Scientific reports, 9(1), 1-10.

Wei, J., Huang, W., Li, Z., Xue, W., Peng, Y., Sun, L., & Cribb, M. (2019). Estimating 1-km-resolution PM2. 5 concentrations across China using the space-time random forest approach. Remote Sensing of Environment, 231, 111221.

Xue, T., Zheng, Y., Tong, D., Zheng, B., Li, X., Zhu, T., & Zhang, Q. (2019). Spatiotemporal continuous estimates of PM2. 5 concentrations in China, 2000–2016: A machine learning method with inputs from satellites, chemical transport model, and ground observations. Environment international, 123, 345-357.

Xue, Y., Li, Y., Guang, J., Tugui, A., She, L., Qin, K., ... & Wang, Z. (2020). Hourly PM2. 5 Estimation over Central and Eastern China Based on Himawari-8 Data. Remote Sensing, 12(5), 855.

Xue, T., Zheng, Y., Geng, G., Zheng, B., Jiang, X., Zhang, Q., & He, K. (2017). Fusing observational, satellite remote

sensing and air quality model simulated data to estimate spatiotemporal variations of PM2. 5 exposure in China. Remote Sensing, 9(3), 221.

Yang, L., Xu, H., & Jin, Z. (2019). Estimating ground-level PM2. 5 over a coastal region of China using satellite AOD and a combined model. Journal of Cleaner Production, 227, 472-482.

Yao, F., Wu, J., Li, W., and Peng, J. (2019). A spatially structured adaptive two-stage model for retrieving ground-level PM2.5 concentrations from VIIRS AOD in China, ISPRS Journal of Photogrammetry and Remote Sensing., 151, 263-276.

Zhang, T., Zang, L., Wan, Y., Wang, W., & Zhang, Y. (2019). Ground-level PM2. 5 estimation over urban agglomerations in China with high spatiotemporal resolution based on Himawari-8. Science of the total environment, 676, 535-544.

**Q2.7: Follow point 3, the authors could try to use this method on other regions where have an extensive ground-based measurements network, for example, North America, to test their validity.**

**Response:** Thank the referee for his/her comment. At present, the study area of our study focuses on China. In the future, we will use this method on other regions, such as North America, to test its validity.

**Technical corrections:**

**Q2.8: Section index is wrong. For example, two "3.2" sections exist in the main text.**

**Response:** Thank the referee for his/her comment. This issue has been fixed in the manuscript.

**Q2.9: Line40: should be "van Donkelaar" instead of "Van Donkelaar".**

**Response:** Thank the referee for his/her comment. This name has been revised in the manuscript.

**The main revision is as follows:**
Hence, the approaches based on Chemical Transport Models (CTMs) (van Donkelaar et al., 2010; Wang et al., 2016; Weagle et al., 2018) or remote sensing satellites (Chen et al., 2018; Li et al., 2020; Stafoggia et al., 2019; Shtein et al., 2020; Wei et al., 2019; Yao et al., 2019; You et al., 2015) have been exploited to enlarge the spatial coverage of the $PM_{2.5}$ and $PM_{10}$ monitoring.

**Q2.10: Define "DUCMASS" in the main text.**

**Response:** Thank the referee for his/her comment. The full name of "DUCMASS" has been provided

in the manuscript.

**The main revision is as follows:**

In the meantime, the rank of DUst Column MASS density (DUCMASS) rises for the estimation of $PM_{10}$ compared to that of $PM_{2.5}$, indicating the flexibility of our approach.